# Doctor Approved: Generating Medically Accurate Skin Disease Images through AI-Expert Feedback

**Janet Wang**[†]   **Yunbei Zhang**[†]   **Zhengming Ding**   **Jihun Hamm**
Tulane University
{swang47, yzhang111, zding1, jhamm3}@tulane.edu

## Abstract

Paucity of medical data severely limits the generalizability of diagnostic ML models, as the full spectrum of disease variability can not be represented by a small clinical dataset. To address this, diffusion models (DMs) have been considered as a promising avenue for synthetic image generation and augmentation. However, they frequently produce *medically inaccurate* images, deteriorating the model performance. Expert domain knowledge is critical for synthesizing images that correctly encode clinical information, especially when data is scarce and quality outweighs quantity. Existing approaches for incorporating human feedback, such as reinforcement learning (RL) and Direct Preference Optimization (DPO), rely on robust reward functions or demand labor-intensive expert evaluations. Recent progress in Multimodal Large Language Models (MLLMs) reveals their strong visual reasoning capabilities, making them adept candidates as evaluators. In this work, we propose a novel framework, coined **MAGIC** (**M**edically **A**ccurate **G**eneration of **I**mages through AI-Expert **C**ollaboration), that synthesizes clinically accurate skin disease images for data augmentation. Our method creatively translates expert-defined criteria into actionable feedback for image synthesis of DMs, significantly improving clinical accuracy while reducing the direct human workload. Experiments demonstrate that our method greatly improves the clinical quality of synthesized skin disease images, with outputs aligning with dermatologist assessments. Additionally, augmenting training data with these synthesized images improves diagnostic accuracy by $+9.02\%$ on a challenging 20-condition skin disease classification task, and by $+13.89\%$ in the few-shot setting. Beyond image synthesis, MAGIC illustrates a task-centric alignment paradigm: instead of adapting MLLMs to niche medical tasks, it adapts tasks to the evaluative strengths of general-purpose MLLMs by decomposing domain knowledge into attribute-level checklists. This design offers a scalable and reliable path for leveraging foundation models in specialized domains. Our implementation detail and code is available at https://github.com/janet-sw/MAGIC.git.

## 1   Introduction

Recent advances in deep learning have made dermatological diagnosis increasingly accessible, offering significant potential for teledermatology in rural regions [6, 15, 37, 52]. However, privacy constraints and proprietary rights over skin images often lead to data scarcity, especially for rare conditions, making it difficult to capture the full complexity and variability of skin diseases for training robust diagnostic models. In response, various data augmentation strategies have been proposed—most straightforwardly, by aggregating open-source dermatological images [1, 60]. Yet,

---

[†]Equal contribution.

39th Conference on Neural Information Processing Systems (NeurIPS 2025).

this approach does not guarantee access to high-quality samples of the precise clinical presentations needed, such as specific combinations of skin tones, body sites, and other lesion characteristics.

Image synthesis by Text-to-Image (T2I) Diffusion Models (DMs) [12] has emerged as a promising solution to enrich datasets under the guidance of prompts. Such controlled generation helps mitigate long-tail distributions, reduce biases against underrepresented groups, and improve model generalization—essential aspects of building reliable diagnostic systems [31, 48, 58, 35]. While the effectiveness of diffusion-based synthetic augmentation for common objects is debatable compared to retrieval-based methods, their value in the medical domain remains significant due to the proprietary nature of medical data and the general infeasibility of retrieval [22]. T2I DMs have been employed to augment medical datasets across various imaging modalities [3, 28, 30, 42, 63, 13]. Previous works have also attempted to fine-tune DMs on skin disease images to enhance subsequent diagnostic model performance. However, these approaches typically involved end-to-end generation without expert participation during the training process, relegating expert assessment or filtering to a post-generation stage, rather than actively guiding the model to create clinically accurate images. [2, 45, 46, 58].

Aligning DMs via Reinforcement Learning from Human Feedback (RLHF) has been explored to adapt these models and generate images that meet human preferences. In particular, [33] proposes reward-weighted likelihood maximization to achieve alignment. Building on this, [53] engages expert pathologists to assess sampled bone marrow images against a clinical plausibility checklist and train a reward function on binary feedback to emulate clinician assessments when fine-tuning a class-conditional DM. More recently, [4, 17] considers the denoising process as a multi-step Markov Decision Process (MDP) and adopts policy gradient optimization to fine-tune DMs based on human feedback. However, such methods still require reliable reward functions, whose training demands substantial computational resources and vast amounts of human-labeled feedback. To address these limitations, [66] proposes using Direct Preference Optimization (DPO) [43], which enables DM fine-tuning directly on preference data, bypassing the need for an explicit reward model and allowing iterative parameter updates based on human feedback at each timestep of the denoising process.

Inspired by recent advances in Reinforcement Learning from AI Feedback (RLAIF) [32] and the strong visual reasoning capabilities of MLLMs, we propose **MAGIC** (**M**edically **A**ccurate **G**eneration of **I**mages through AI-Expert **C**ollaboration), a semi-automated framework that utilizes MLLMs for visual evaluation. In this framework, human experts are primarily required to: (1) craft, from credible sources, checklists that are easily verifiable by a MLLM, and (2) oversee the MLLM's feedback on synthetic images during the training of T2I DMs. By iteratively learning from the feedback enhanced with expert knowledge, MAGIC steers the T2I DMs toward more medically consistent generations. This approach highlights the potential of AI-expert collaboration, as MAGIC effectively leverages existing domain knowledge without labor-intensive annotation. Moreover, MAGIC incorporates an Image-to-Image (I2I) module within its training pipeline to initiate denoising from intermediate timesteps rather than pure Gaussian noise. This accelerates the sampling stage while ensuring factorized lesion transformations that do not deviate excessively from the real data distribution.

Through rigorous experiments, we demonstrate that our MAGIC framework performs effectively with both reward-based fine-tuning (RFT) and DPO, exhibiting particular strength with DPO. The MAGIC-DPO pipeline optimizes DMs to generate synthetic data that accurately represent each condition's unique visual features, with improvements observed as training progresses and more image-feedback pairs are used (Fig. 2). This is also validated by increasing dermatologist evaluation scores (Fig. 4d) and decreasing Fréchet Inception Distance (FID) scores (Fig. 4c), indicating improved clinical accuracy and fidelity. As a result, we also observe significant improvements in classification performance over baseline, highlighting MAGIC's potential to advance AI dermatology. Overall, our main contributions are: **(i)** We propose **MAGIC**, a novel fine-tuning framework that integrates expert knowledge into DMs, enabling their subsequent fine-tuning with both DPO and RFT. The framework incorporates an I2I module to efficiently align the model for producing medically accurate images. **(ii)** Our framework employs an AI-Expert collaboration paradigm that offloads the work of visual evaluation to a powerful MLLM under minimal expert supervision, significantly reducing time and labor required from medical experts. **(iii)** MAGIC, particularly when combined with DPO (MAGIC-DPO), generates high-quality, clinically accurate images, achieving notable improvements in FID scores and classification performance. It yields a $+9.02\%$ boost in accuracy on a challenging 20-condition classification task and a $+13.89\%$ improvement in few-shot scenarios.

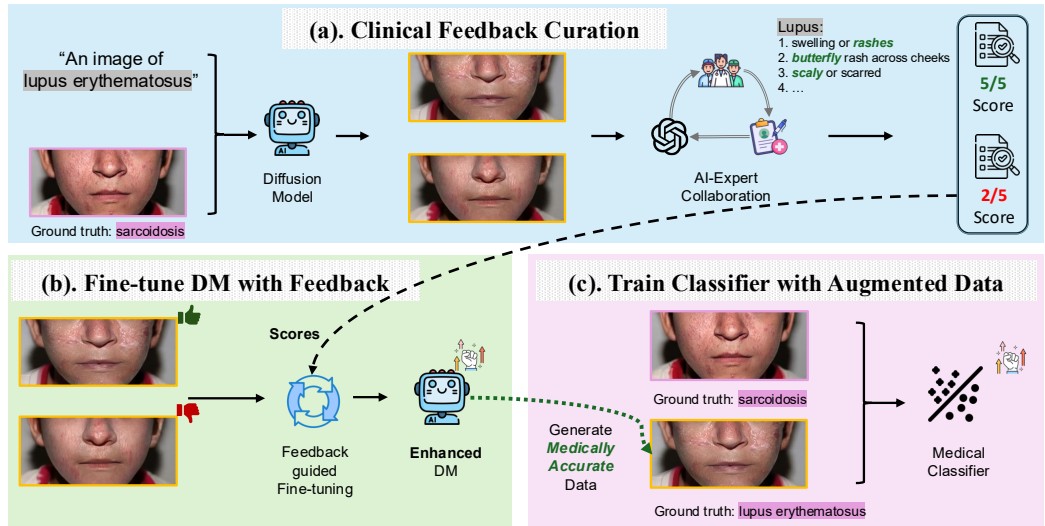

Figure 1: Illustration of our proposed **MAGIC**: (a) A preliminary fine-tuned diffusion model (DM) transforms a source image (e.g., sarcoidosis) to a target condition (e.g., lupus erythematosus); an MLLM then provides expert checklist-based feedback scores on the generated image pair. (b) This feedback guides the subsequent fine-tuning (e.g., RFT or DPO) of the DM. (c) The feedback-enhanced DM synthesizes medically accurate dermatological images for robust classifier training.

## 2   Related Works

**DM-based Augmentation for Skin Disease Classification.** Existing studies have explored diffusion models (DMs) to generate synthetic dermatological images for augmenting the training data of diagnostic models. Along this line, [46] implemented a seed-based approach, sampling a small set of real images from the Fitzpatrick17k dataset [23] and generating synthetic data using the inpainting feature of OpenAI's DALL·E 2. Subsequently, [45] leveraged Stable Diffusion's T2I pipeline, fine-tuned with Dreambooth, to produce images of specific disease conditions. Other related works [2, 31] have similarly employed DM-based augmentation to enhance diagnostic accuracy and generalization on their internal skin disease datasets. Building on these advances, [58] proposed a diffusion augmentation framework specifically targeting minority skin types. Their approach involved Textual Inversion [19] and Low-Rank Adaptation (LoRA) [26] for fine-tuning, coupled with image-to-image generation for inference. This method enabled the creation of images depicting novel lesion concepts previously unseen by the DM. Their study revealed that images synthesized using this dual-guidance strategy improved the diagnostic performance of subsequent classifiers for minority skin types, even when reference data from these groups was absent from the training set. However, expert involvement in these previously proposed methods, if any, is typically confined to post-generation assessment or filtering, rather than actively guiding the image creation process.

**Fine-tune Diffusion Models (DMs) with Feedback.** Approaches to fine-tuning DMs with human feedback broadly fall into two categories: reward-based and preference-based. Reward-based methods [5, 16, 18, 34, 65] depend on robust reward models, the training of which typically requires substantial datasets and extensive human evaluations. In the medical domain, for instance, [53] leveraged reward-weighted maximization to synthesize plausible bone marrow images, by fine-tuning a class-conditional DM with a pathologist's feedback on synthetic images. In contrast, preference-based approaches aim to derive policies directly from preference data, thereby bypassing the need for explicit reward functions [9, 14, 32]. A key development in this area is Direct Preference Optimization (DPO) [43], originally proposed for fine-tuning language models directly using preferences. While DPO adaptations for diffusion models have primarily been tested for image-feedback alignment [57, 66], their application to medical image generation remains largely unexplored, especially for clinical images of skin diseases, which exhibit high complexity and variations.

**MLLMs-as-a-Judge.** Collecting high-quality feedback has traditionally relied on human labelers, an approach that is both costly and difficult to scale. Recent research demonstrates that powerful

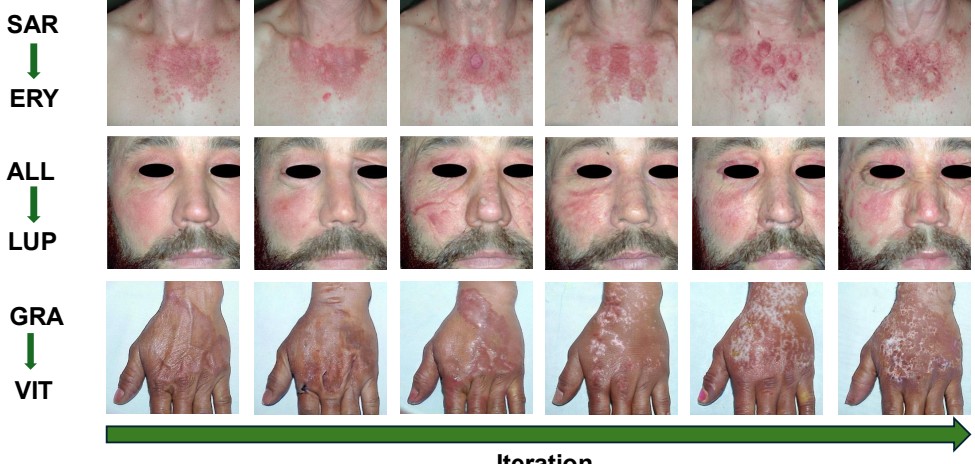

**Iteration**

Figure 2: Evolution of synthetic skin conditions generated by MAGIC-DPO, illustrating its ability to learn unique visual features from feedback across training iterations. The Top Row demonstrates the model transforming Sarcoidosis (SAR) into Erythema Multiforme (ERY), learning features like target (bull's-eye) lesions with concentric rings. The Middle Row demonstrates the model transforming Allergic Contact Dermatitis (ALL) into Lupus Erythematosus (LUP), progressively developing a butterfly rash covering the cheeks. The Bottom Row demonstrates the model transforming Granuloma Annulare (GRA) into Vitiligo (VIT), evolving to show characteristic depigmented patches.

.

proprietary MLLMs, such as GPT-4V and GPT-4o [40], can serve as effective generalist evaluators for vision-language tasks [8, 21, 69]. These models have proven particularly valuable in complex tasks requiring human-like judgment, including visual conversations and detailed image captioning, where MLLMs are often incorporated into evaluation benchmarks to assess model responses [54, 68, 72]. More recently, these models have shown capabilities in encoding clinical knowledge and acting as evaluators in medical reasoning [51]. Although employing MLLMs as collaborators in AI dermatology holds great potential to enhance the reliability of diagnostic models, the optimal paradigm for their collaboration with medical experts still remains underexplored.

## 3 Method

### 3.1 Preliminaries

**Diffusion Models (DMs).** DMs are designed to learn the probability distribution $p(x)$ by reversing a Markovian forward process, denoted as $q(\boldsymbol{x}_t \mid \boldsymbol{x}_{t-1})$, which incrementally introduces noise into the images. The reversal, a denoising process, is implemented through a neural network tasked with predicting either the mean of $\boldsymbol{x}_{t-1}$ or the noise $\boldsymbol{\epsilon}_{t-1}$ from the forward process. In our approach, we utilize a network $\boldsymbol{\mu}_\theta(\boldsymbol{x}_t; t)$ to predict the mean of $\boldsymbol{x}_{t-1}$, rather than the added noise. We employ the Mean Squared Error (MSE) as a performance metric, defining the objective function of our network as follows:

$$\mathcal{L}_{\mathrm{DM}} = \mathbb{E}_{t \sim [1,T], \boldsymbol{x}_0 \sim p(\boldsymbol{x}_0), \boldsymbol{x}_t \sim q(\boldsymbol{x}_t|\boldsymbol{x}_0)} \left[ \|\tilde{\boldsymbol{\mu}}(\boldsymbol{x}_0, \boldsymbol{x}_t) - \boldsymbol{\mu}_\theta(\boldsymbol{x}_t, t)\|^2 \right], \tag{1}$$

where $\tilde{\boldsymbol{\mu}}_\theta(\boldsymbol{x}_t, \boldsymbol{x}_0)$ represents the posterior mean of the forward process.

In conditional generative modeling, diffusion models are adapted to learn the conditional distribution $p(x|\boldsymbol{c})$, where $\boldsymbol{c}$ represents conditioning information, such as image categories or captions. This adaptation involves augmenting the denoising network with additional input, $\boldsymbol{c}$, resulting in $\boldsymbol{\mu}_\theta(\boldsymbol{x}_t, t; \boldsymbol{c})$. To generate a sample from the learned distribution $p_\theta(x|\boldsymbol{c})$, we initiate the process by drawing a sample $\boldsymbol{x}_T \sim \mathcal{N}(\boldsymbol{0}, \mathbf{I})$, which is then progressively denoised through iterative application of $\boldsymbol{\epsilon}_\theta$, based on specific samplers adopted [25]. The reverse process is modeled as:

$$p_\theta(\boldsymbol{x}_{t-1} \mid \boldsymbol{x}_t, \boldsymbol{c}) = \mathcal{N}\left(\boldsymbol{x}_{t-1}; \boldsymbol{\mu}_\theta(\boldsymbol{x}_t, \boldsymbol{c}, t), \sigma_t^2 \mathbf{I}\right). \tag{2}$$

In our skin disease image generation framework, we leverage the I2I pipeline of Stable Diffusion [44] to transform lesion features while preserving body part information in the image. This strategy effectively reduces semantic distortion during generation and ensures factorized translation of lesions, thereby enhancing medical plausibility. Specifically, we start with a real input dermatological image $x_0$ (e.g., sarcoidosis), add partial noise to it, and transform it into a different target skin condition (e.g., lupus erythematosus), by denoising this partly noised images. And the denoising process is governed by $\mu_\theta$ and denoise strength parameter $\gamma$.

**Multi-Step MDP Formulation.** We formulate the diffusion model's denoising process as a multi-step Markov Decision Process (MDP), following [5, 55]. In our model, the state $s \in \mathcal{S}$ includes the current denoising time step, denoised image data and prompt. The action space $\mathcal{A}$ includes possible image transformations at each time step. The state transition function $P(s'|s, a)$ describes the image evolution, and the reward function $r(s, a)$ assigns values based on the image quality at each time step, aiming to maximize cumulative returns $\mathcal{J}(\pi) = \mathbb{E}_\tau[\sum_{t=0}^{T-1} r(s_t, a_t)]$. The MDP is formulated as

$$
\begin{aligned}
\mathbf{s}_t &\triangleq (\boldsymbol{c}, t, \boldsymbol{x}_{T-t}), \quad P(\mathbf{s}_{t+1} \mid \mathbf{s}_t, \mathbf{a}_t) \triangleq (\delta_{\boldsymbol{c}}, \delta_{t+1}, \delta_{\boldsymbol{x}_{T-1-t}}); \\
\mathbf{a}_t &\triangleq \boldsymbol{x}_{T-1-t}, \qquad \pi(\mathbf{a}_t \mid \mathbf{s}_t) \triangleq p_\theta(\boldsymbol{x}_{T-1-t} \mid \boldsymbol{c}, t, \boldsymbol{x}_{T-t}); \\
\rho_0(\mathbf{s}_0) &\triangleq (p(\boldsymbol{c}), \delta_0, \mathcal{N}(\mathbf{0}, \mathbf{I})); \\
r(\mathbf{s}_t, \mathbf{a}_t) &\triangleq r((\boldsymbol{c}, t, \boldsymbol{x}_{T-t}), \boldsymbol{x}_{T-t-1}),
\end{aligned}
\tag{3}
$$

where $\delta_x$ represents the Dirac delta distribution, and $T$ denotes the maximize denoising timesteps.

## 3.2 Preliminary Diffusion Models Fine-tuning

Previous studies have shown that off-the-shelf diffusion models struggle to represent skin lesion concepts, making preliminary fine-tuning necessary before aligning with expert feedback [58]. Following [58], we employ Latent Diffusion Models (LDMs) [44], which operate in autoencoder latent space to reduce computational demands while maintaining generation quality. For simplicity, we abuse notation and use $x$ to represent the latent input to the diffusion process rather than the original image. Our framework utilizes Textual Inversion [20] to derive unique embeddings that capture the semantics of each condition extracted from training data. Each image is paired with a descriptive string containing placeholders (e.g., 'an image of $\{S_*\}$') as input. The optimal embedding $v_*$, encapsulating the lesion concept $S_*$, is then obtained by minimizing reconstruction loss while keeping the LDM fixed. To enhance the efficiency of the LDM fine-tuning process, we employ LoRA [26], adapting the model with the discovered tokens from Textual Inversion. This approach maintains the pre-trained model weights while introducing only two compact matrices $A$ and $B$ (where $A \in \mathbb{R}^{n \times r}, B \in \mathbb{R}^{r \times n}$). These matrices are embedded within the attention layers, enabling the detailed capture of skin lesion characteristics previously unrepresented in the initial model, aligned with the learned target embedding $v_*$.

## 3.3 Expert Feedback Curation

While diffusion models can synthesize visually realistic medical images, their clinical validity often remains questionable [53]. Incorporating medical expertise is therefore crucial for guiding these models to generate medically accurate images. To provide this clinical guidance, our framework leverages structured feedback derived from checklists that are designed by an experienced dermatologist. These checklists evaluates five distinct aspects of each condition: [Location, Lesion Type, Shape/Size, Color, Texture] (see Appendix B for complete details). Assessment against these aspects yields a binary outcome (e.g., satisfied/not satisfied) for each criterion. To automate this evaluation, we instructed an MLLM to analyze each synthesized image based on the target condition's checklist and return a 5-dimensional binary score list, where each dimension corresponds to a criterion's satisfaction (see Appendix C for instruction details). To accommodate both reward-based and preference-based alignment strategies, we generate a pair of images from each text prompt and submit each single image to the MLLM for this assessment. Thus, the MLLM's score list for each image in a pair individually stands as a sample for RFT, while the pair of score lists can be used for DPO. Examples of this MLLM assessment using OpenAI's GPT-4o are illustrated in Fig. 3, showing yielded score lists such as [1,0,0,1,0] and [1,1,1,1,1] for a given pair. Ultimately, each 5-dimensional MLLM-generated score list is aggregated into an overall binary score (e.g., 0 for negative example, 1 for positive example) using a predefined algorithm (detailed in Appendix A.2). This semi-automated pipeline allows us to significantly accelerate the curation of expert feedback. Notably, only synthetic images are sent to GPT-4o API services and no real patient images are processed by the MLLM, to preserve privacy.

## 3.4 Finetuning with Expert Feedback

After collecting pairwise preferences, we explore two complementary ways to integrate them into optimizing the diffusion model parameters $\theta$.

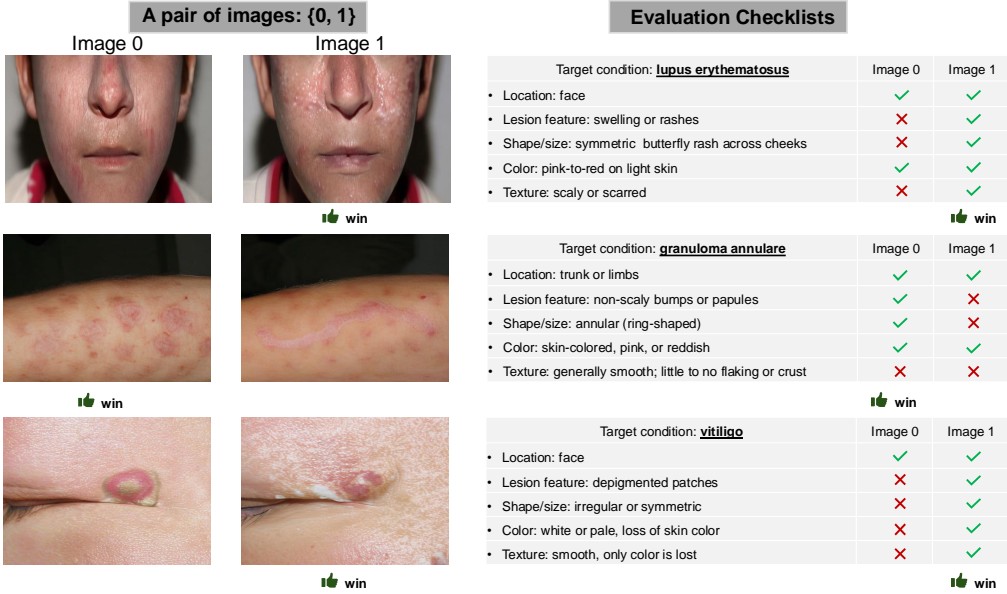

Figure 3: Illustration of the image assessment process by OpenAI's GPT-4o using condition-specific checklists for target skin conditions such as lupus erythematosus, granuloma annulare, and vitiligo. Each generated image in a pair is evaluated against five clinical criteria. The image with more satisfied criteria is considered the preferred sample in a comparison. Additional examples are in Appendix 7.

**Reward-model guided fine-tuning (RFT)** Let $\mathcal{R}_\phi : \mathbb{R}^{H\times W\times 3} \times \mathcal{C} \to \mathbb{R}_{\geq 0}$ be a learned scalar that predicts the likelihood an image $x$ conditioned on class $c$ satisfies every checklist item. We follow [33, 53] and mix real and synthetic images when training $\mathcal{R}_\phi$ with an MSE loss. Formally, with feedback labels $y \in \{0,1\}$ we minimize $\mathcal{L}_{\mathrm{RM}}(\phi) = \sum_{(x,c,y)}\big(y - \mathcal{R}_\phi(x,c)\big)^2$. After fitting $\phi$, we refine $\theta$ by maximising the expected reward-weighted log-probability of the action sequence generated along each denoising trajectory $\sigma = \{(s_t, a_t)\}_{t=0}^{T-1}$:

$$\mathcal{L}_{\mathrm{RFT}}(\theta) = \mathbb{E}_{(x,c)\sim\mathcal{D}_{\mathrm{s}}}\Big[-\mathcal{R}_\phi(x,c)\textstyle\sum_{t=0}^{T-1}\log\pi_\theta(a_t \mid s_t)\Big] + \beta_r\,\mathbb{E}_{(x,c)\sim\mathcal{D}_{\mathrm{r}}}\Big[-\textstyle\sum_{t=0}^{T-1}\log\pi_\theta(a_t \mid s_t)\Big], \qquad (4)$$

where $\mathcal{D}_{\mathrm{s}}$ and $\mathcal{D}_{\mathrm{r}}$ denote synthetic and real image pools, respectively, and $\beta_r$ balances fidelity to expert feedback against faithfulness to the original data distribution.

**Direct Preference Optimization (DPO)** Given a pair of trajectories $(\sigma^w, \sigma^l)$ that yield a *winner* image $x^w$ and a *loser* image $x^l$ under expert comparison, DPO increases the likelihood of every action $a_i^w$ on the winning branch while decreasing the likelihood of the corresponding $a_i^l$ on the losing branch. Similar to reinforcement learning methods [7, 49, 50], rewards are assigned by $\forall s_t, a_t \in \sigma, r(s_t, a_t) = 1$ for winning the game and $\forall t \in \sigma, r(s_t, a_t) = -1$ for losing the game. Following [66], we also assume that if the final image is preferred, then any state-action pair in its generation path is superior to the corresponding pair in the non-preferred path. To maximize learning from each generation process under this assumption, we construct $t' = \gamma T$ sub-segments that allow the model to learn from intermediate states

$$\mathcal{L}_{\mathrm{DPO}}^i(\theta) = -E_{(s_i,\sigma_w,\sigma_l)}\Big[\log\rho\big(\beta\log\tfrac{\pi_\theta(a_i^w|s_i^w)}{\pi_{\mathrm{ref}}(a_i^w|s_i^w)} - \beta\log\tfrac{\pi_\theta(a_i^l|s_i^l)}{\pi_{\mathrm{ref}}(a_i^l|s_i^l)}\big)\Big], \qquad (5)$$

where $i \in [0, t'-1]$, effectively increasing data utilization by a factor of $t'$.

### 3.5   Synthetic Augmentation for Classifier Training

After fine-tuning a diffusion model with expert-enhanced feedback, we leverage the model to synthesize images for dataset augmentation, primarily through an image-to-image translation approach. For any given real sample $x$ with label $y$, we first randomly select a different target label $y'$ from the label set. We then use the text prompt "an image of $\{y'\}$"—incorporating the specific text embedding for $y'$ learned via Textual Inversion—to guide the DM in generating a new image $x'$. This process is designed so that $x'$ preserves most of the anatomical context of the original sample $x$ while primarily displaying the lesion semantics of the target label $y'$, thereby achieving a factorized transformation. This I2I generation strategy offers a key benefit: it helps mitigate the risk of the classifier learning spurious correlations by preventing it from associating lesions with specific body locations, encouraging a focus on the intrinsic characteristics of the skin lesions. During the subsequent classifier training phase, we intentionally control the influence of synthetic data using a ratio parameter $\rho \in (0,1)$,

which determines the percentage of synthetic images added to each training batch. While our method aims to generate medically accurate images, potential domain shifts between real and synthetic data remain an important consideration. Indeed, our experiments indicate that varying the proportion of synthetic data can significantly affect classifier performance on real test data (see Fig. 4a).

## 4 Experiments

**Dataset.** Following prior work [58], we use the Fitzpatrick17k dataset to evaluate our synthetic augmentation pipeline [23]. Fitzpatrick17k contains clinical photos of 114 skin conditions, each annotated with a condition label and a Fitzpatrick Skin Type (FST). Although there are other datasets of clinical photos (e.g., SCIN [62] and DDI[11]), they are primarily collected within the United States and feature lighter skin tones. Fitzpatrick17k encompasses a wider range of skin types, making it particularly suitable for evaluating generalizable diagnostic approaches. For our experiments, we focus on a subset of the Fitzpatrick17k dataset consisting of 20 skin conditions. We chose these based on two criteria: (1) they present the largest class sizes in the dataset, and (2) they have well-established descriptions available from reputable clinical sources (e.g., Mayo Clinic, Cleveland Clinic), which allowed dermatologists to craft reliable diagnostic checklists of key visual features for these diseases. These checklists, verified by clinicians, distill essential visual cues for each condition, detailed in the Appendix B. The distribution of the selected classes is provided in the Appendix A.

**Models and Baselines.** We utilize Stable Diffusion v2-1 [44] for image generation. For classification tasks, we employ ResNet18 [24] and DINOv2 [41] as backbone architectures. For medical image generation, we evaluate four different methods: (1) diffusion model fine-tuned with Textual Inversion and LoRA, generating images via text-to-image (+ T2I); (2) the same fine-tuned model but generating via image-to-image (+ I2I); and (3/4) our proposed MAGIC (RFT/DPO) with expert feedback. We assess synthetic image quality using both FID score and human evaluation. For classification experiments, we first establish a baseline by training a classifier solely on real data. We then generate an equivalent number of synthetic images using each generation method (excluding the off-the-shelf DM due to its lack of domain-specific knowledge [58]), and train classifiers on combined real and synthetic datasets. Implementation details are provided in Appendix A.

**Implementation Details.** To adapt the model to skin lesion concepts, our preliminary fine-tuning process proceeds in two stages: (i) We learn unique disease-related tokens by updating the text encoder via Textual Inversion [19], thereby introducing new vocabulary specific to each condition; and (ii) we tie the newly learned tokens to fine-grained visual cues within the images by updating the UNet parameters via LoRA [26]. Further details on prompts and hyperparameters can be found in the Appendix A.

For training with expert feedback, all experiments share a unified *sampling-feedback* pipeline. For each mini-batch of image-prompt pairs drawn from the real set, the current diffusion model generates two synthetic variants via the Stable-Diffusion image-to-image path, intentionally targeting skin-disease classes that differ from the originals to maximise diversity. Each synthetic image is then scored with the condition-specific checklists (Appendix B), which we submit to GPT-4o [40]. The API returns binary vectors indicating whether each criterion is met; if the lesion is deemed invalid, an all-zero vector is assigned. From every pair of vectors we derive a *winner-loser* label and store the associated latents, timesteps, and prompt embeddings. We subsequently branch into two finetuning regimes: (i) in the *reward-model route* we fit a scalar network $\mathcal{R}_\phi$ to these binary outcomes and update $\theta$ by the reward-weighted likelihood of Eq. (4); (ii) in the *DPO route* we treat each preference tuple as in [66] and optimize the multi-segment loss of Eq. (5). Both routes draw from the same pool of feedback pairs, subsequent comparisons isolate the effect of the finetuning algorithm itself. Examples are visualised in Fig. 3.

For classifier training, we randomly split the dataset into training and hold-out sets at a 50/50 ratio, resulting in 3,100 training and 3,100 test images. The baseline classifier is trained exclusively on this 3,100-image training set. During inference, we apply the same hyperparameters used in the DPO sampling stage when generating synthetic images with the DPO fine-tuned model. We generate one synthetic image for each real image, intentionally assigning a target label that differs from the real image's original label while corresponding to the same body region. Following established practices, we combine synthetic and real images to optimize performance, maintaining a fixed ratio of synthetic to real examples in each training batch. All experiments are conducted *five* rounds on RTX 6000 Ada GPUs. Our experimental evaluation encompasses both CNN-based and Transformer-based classifier architectures, fine-tuned according to protocols outlined in previous work [58].

## 5 Analysis

### 5.1 Experimental Results

**Classification results.** We comprehensively evaluate synthetic image quality by its impact on downstream classification using ResNet18 and DINOv2 architectures (Tables 1 and 2). Our MAGIC framework markedly enhances performance across both models compared to baselines. Standard fine-tuned Text-to-Image (T2I) generation degrades ResNet18 accuracy by $-3.74\%$ and DINOv2 by $-2.16\%$, while the fine-tuned Image-to-

Table 1: Performance of ResNet18-based classifiers trained on real and synthetic data.

| Method | Acc | F1 | Prec | Rec |
|---|---|---|---|---|
| Real | 29.31 | 28.73 | 28.61 | 29.13 |
| + T2I | 25.57 | 24.63 | 24.44 | 25.16 |
|  | -3.74 | -4.11 | -4.17 | -3.97 |
| + I2I | 31.45 | 31.09 | 31.03 | 31.49 |
|  | +2.14 | +2.35 | +2.42 | +2.36 |
| + MAGIC | 33.49 | 30.40 | 29.12 | 29.67 |
| (RFT) | +4.18 | +1.67 | +0.51 | +0.54 |
| + MAGIC | **38.33** | **37.01** | **38.41** | **36.06** |
| (DPO) | **+9.02** | **+8.28** | **+9.80** | **+6.94** |

Table 2: Performance of DINOv2-based classifiers trained on real and synthetic data.

| Method | Acc | F1 | Prec | Rec |
|---|---|---|---|---|
| Real | 49.89 | 49.43 | 50.03 | 49.31 |
| + T2I | 47.73 | 47.26 | 47.51 | 47.43 |
|  | -2.16 | -2.17 | -2.52 | -1.88 |
| + I2I | 50.71 | 50.17 | 51.04 | 49.89 |
|  | +0.82 | +0.74 | +1.01 | +0.58 |
| + MAGIC | 51.16 | 52.66 | 52.17 | 52.69 |
| (RFT) | +1.27 | +3.23 | +2.14 | +3.38 |
| + MAGIC | **55.01** | **54.05** | **54.96** | **53.70** |
| (DPO) | **+5.12** | **+4.62** | **+4.93** | **+4.39** |

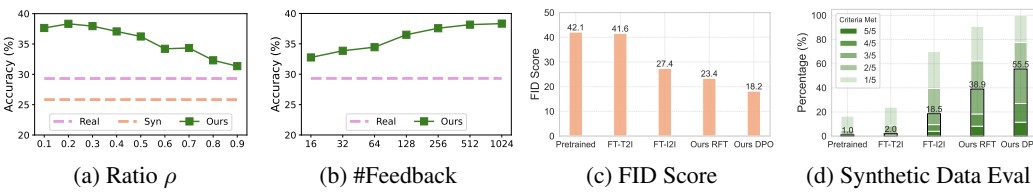

(a) Ratio $\rho$     (b) #Feedback     (c) FID Score     (d) Synthetic Data Eval

Figure 4: Experimental results showing (a) the impact of ratio $\rho$, (b) feedback volume on accuracy, (c) FID score comparison across different methods, and (d) evaluation results on synthetic data showing the percentage of criteria met. Our method consistently outperforms baseline methods in most metrics, achieving lower FID scores and higher criteria satisfaction rates.

Image (I2I) approach offers modest gains, increasing ResNet18 accuracy by $+2.14\%$ and DINOv2 by $+0.82\%$. The feedback integrated via our MAGIC framework proves beneficial for both Reward-model guided Fine-Tuning (RFT) and Direct Preference Optimization (DPO) strategies. Specifically, MAGIC-RFT improved accuracy over the real data baseline by $+4.18\%$ for ResNet18 and $+2.21\%$ for DINOv2. MAGIC-DPO demonstrated even more substantial gains, boosting accuracy by $+9.02\%$ for ResNet18 (from 29.31% to 38.33%) and by $+5.12\%$ for DINOv2 (from 49.89% to 55.01%), with similar improvements in F1, precision, and recall. We further validate the MAGIC framework on additional datasets, SCIN and PAD-UFES-20, with results in Appendix D.3.

The DPO approach within the MAGIC framework (MAGIC-DPO) shows particular strength. Its advantage may stem from directly optimizing for preference alignment without an intermediate reward model. This can be more robust and generalize better, proving especially advantageous when the number of feedback pairs is limited, as is common in specialized medical domains, thus sidestepping potential instabilities in reward modeling. The quality of expert guidance remains crucial for generating synthetic images that are not only visually plausible but also encode clinically relevant diagnostic features. This enhanced alignment is reflected across our evaluations, including improved qualitative outputs (Fig. 2), FID scores (Fig. 4c), and expert preference measures (Fig. 4d). We also evaluate MAGIC-DPO with other classifier backbones (see Appendix D.5 for details).

**Expert evaluation on generated images.** To further assess the quality and medical plausibility of images generated by our methods, we engaged medical experts to evaluate the synthetic data based on our specific checklist criteria. For each method, we sampled 10 images per skin condition, resulting in 200 images per method. Each image was evaluated against 5 criteria, with binary outcomes (satisfied/not satisfied). Fig. 4d summarizes these evaluation results, displaying the percentage of images meeting different numbers of criteria (with details in Appendix B). The results show that images from the pretrained diffusion model rarely satisfied more than one criterion, and none met more than three. Standard Text-to-Image (T2I) generation showed minimal improvement, with only 2.0% of images meeting 3 or more criteria and only a single image meeting 4 criteria overall. Fine-tuned Image-to-Image (I2I) generation yielded better outputs, with 18.5% of its images meeting 3 or more criteria, underscoring I2I's greater suitability for medical tasks. Our MAGIC framework significantly builds on this; MAGIC-RFT (Ours RFT) further increased the proportion of high-quality images, with 38.9% meeting 3 or more criteria. Notably, MAGIC-DPO (Ours DPO) demonstrated the best performance, with 55.5% of its images satisfying 3 or more criteria. This substantial improvement over both fine-tuned I2I and MAGIC-RFT correlates directly with the observed enhancements in classifier performance.

Table 3: Performance of DINOv2-based classifiers in *few-shot* setting.

| Method | Acc | F1 | Prec | Rec |
|---|---|---|---|---|
| Real (all) | 49.89 | 49.43 | 50.03 | 49.31 |
| Real (310) | 26.45 | 19.50 | 21.86 | 20.19 |
| + T2I | 25.58 | 19.58 | 20.87 | 19.27 |
|  | -2.17 | +0.08 | -0.99 | -0.92 |
| + I2I | 30.10 | 27.26 | 28.07 | 27.00 |
|  | +3.65 | +7.76 | +6.21 | +6.81 |
| + MAGIC | 37.39 | 36.90 | 37.95 | 36.94 |
| (DPO) | +10.94 | +17.40 | +16.09 | +16.75 |
| + MAGIC-A | 40.34 | 39.43 | 42.20 | 38.77 |
| (DPO) | +13.89 | +19.93 | +20.34 | +18.58 |

Table 4: Performance of classifiers across different backbones and Coarse/Structured checklists.

| Model | Method | Acc | F1 | Prec | Rec |
|---|---|---|---|---|---|
| RN18 | Real | 29.31 | 28.73 | 28.61 | 29.13 |
|  | + MAGIC | 32.83 | 30.58 | 29.75 | 31.18 |
|  | Coarse | +3.52 | +1.85 | +1.14 | +2.05 |
|  | + MAGIC | 38.33 | 37.01 | 38.41 | 36.06 |
|  | Structured | +9.02 | +8.28 | +9.80 | +6.94 |
| DINO | Real | 49.89 | 49.43 | 50.03 | 49.31 |
|  | + MAGIC | 51.16 | 52.66 | 52.17 | 52.69 |
|  | Coarse | +1.27 | +3.23 | +2.14 | +3.38 |
|  | + MAGIC | 55.01 | 54.05 | 54.96 | 53.70 |
|  | Structured | +5.12 | +4.62 | +4.93 | +4.39 |

**Few-shot Setting.** We further evaluate our framework in a few-shot setting where only a small number of labeled data are available. This scenario better reflects real-world conditions, as collecting and labeling medical data is both challenging and expensive. We simulate this setting by randomly selecting 10% of the DINOv2 training set (310 images) while keeping the test set fixed. We fine-tuned the diffusion model on these 310 real images using our DPO-based approach (MAGIC-DPO) and other baselines. As shown in Table 3, MAGIC-DPO improves classifier accuracy by +10.94% (from 26.45% to 37.39%) compared to training with only the limited real data, significantly outperforming standard T2I and I2I augmentation baselines in this data-scarce context. Moreover, in practical scenarios, unlabeled medical data from the same distribution may be available even when expert labeling is cost-prohibitive. Our MAGIC framework can effectively utilize such *unlabeled* data; specifically, during the DPO fine-tuning stage, unlabeled data is processed by the diffusion model with randomly selected skin conditions, and feedback is evaluated solely based on the target condition. This makes our framework well-suited for leveraging unlabeled data. This augmented approach, termed **MAGIC-A** (also DPO-based), demonstrates that by incorporating an equal number of unlabeled samples (310), we can further improve accuracy by an additional 2.95% over MAGIC-DPO, reaching 40.34% accuracy.

**Hallucination-Resistant by Design.** The MAGIC framework is explicitly designed to minimize the risk of MLLM hallucination through our AI-Expert collaboration paradigm. The MLLM is not asked to perform open-ended reasoning. Instead, its role is constrained to evaluating an image against a predefined clinical checklist. These checklists, designed by dermatologists, decompose complex medical concepts into simple, visually verifiable features. For instance, while an MLLM may not intrinsically understand "Lupus Erythematosus," it can effectively verify "symmetric butterfly rash across the cheeks". This approach transforms a complex diagnostic reasoning task into a series of closed-question evaluations, which are far less susceptible to unconstrained hallucination. Even so, an analysis of GPT-4o's hallucination is important. To that end, we have GPT-4o evaluate 100 real images from an internal dataset with clinical records confirmed by in-person visits. The MLLM is tasked with describing each image based on the five criteria and a dermatologist assess these descriptions. As shown in Fig. 5, the dermatologist assigned an alignment score of 3 or greater (on a 5-point scale) to approximately 86% of the image-description pairs. Notably, many of the lower-scoring examples were also identified by the dermatologist as being visually ambiguous and challenging for a human to assess from an image alone.

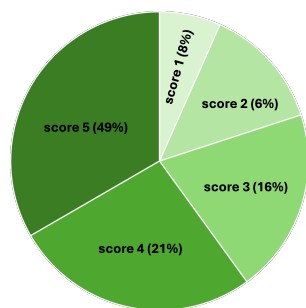

Figure 5: Distribution of alignment scores indicating the number of checklist criteria met in the description.

## 5.2 Abaltion Study

**Effect of Checklist Quality.** We investigate the impact of checklist detail level on the MAGIC framework's efficacy in DPO training, by comparing two types of expert-designed checklists: a "Coarse" version using single-sentence descriptions for each condition, and a more detailed, "Structured" version (as used throughout the main paper and detailed in Appendix B). Table 4 shows that the quality of the checklist is crucial to feedback quality. For the ResNet18 (RN18), augmenting with MAGIC-DPO using Coarse checklists improved accuracy by +3.52% over the real data baseline (from 29.31% to 32.83%), whereas Structured checklists led to a much larger gain of +9.02% (to 38.33%). A similar trend was observed with the DINOv2 (DINO): Coarse checklists yielded a +1.27% accuracy improvement (from 49.89% to 51.16%), while Structured checklists achieved a +5.12% boost (to 55.01%). These results underscore that more detailed and well-structured expert guidance in the checklists significantly enhances the quality of synthetic images and subsequent classifier performance. To explore the upper bounds of this effect, we have a dermatologist craft even more fine-grained checklists with nine criteria (Location, Distribution, Lesion Type, Shape/Size, Border, Elevation, Texture, Color, and

Translucency/Content). We then use these 9-criteria checklists in our MAGIC framework. We observed a small, additional improvement over our original 5-criteria structured checklist, as detailed in the Appendix D.6.

**Effect of feedback volume.** In addition to feedback quality, we also investigate how the quantity of feedback influences image quality and classifier performance. As DPO training progresses, more image pairs are used, providing additional feedback to guide the diffusion model. We visually demonstrate the evolution of generated images across epochs in Fig. 2. Additionally, we also train classifiers using synthetic data that is generated from different training stages. Results in Fig. 4b show that accuracy consistently improves as DPO training accumulates more feedback, with performance stabilizing after receiving feedback from approximately 512 image pairs. Based on these findings, we fix the feedback volume at 1024 image pairs for all our experiments.

**Effect of the ratio $\rho$ of synthetic data.** We investigate how the ratio $\rho$ of synthetic data affects classifier performance. Initial experiments with purely synthetic data failed to achieve performance comparable to real data-trained classifiers. It's expected that, without the guidance of real data, classifiers tend to overfit to the synthetic data distribution. We therefore systematically controlled the percentage of synthetic data used in each training batch across different values of $\rho$, while keeping the total volume of synthetic data constant. As shown in Fig. 4a, performance improves when $\rho$ is less than 0.5 (when synthetic data constitutes less than half of the training data). The performance remains stable when $\rho \in [0.1, 0.3]$. We adopt $\rho = 0.2$ for all our experiments.

**Choice of MLLM.** The MAGIC framework is designed to be model-agnostic, allowing for the flexibility to use the most suitable MLLM for a given task. Therefore, we also test our MAGIC framework with an open-source MLLM, Google DeepMind's MedGemma-4B [47], a foundation model for medical text and image comprehension. We used MedGemma as an evaluator to assess the same set of 1,024 pairs of synthesized images that GPT-4o had assessed. As shown in Table 5, we observed that MedGemma is comparable to GPT-4o as an evaluator in aligning the DM using the MAGIC framework. This demonstrates the flexibility of our MAGIC framework, which is adaptive to both large, closed-source, generalist models and smaller, open-source, domain-specific alternatives. Notably, we observe a small performance gap between the pipelines

Table 5: Performance of linear classifiers trained on synthetic data from the MAGIC pipeline, aligned using feedback from different MLLMs.

| MLLM | Model | Acc | F1 | Prec | Rec |
|------|-------|-----|-----|------|-----|
| MedGemma-4B | ResNet-18 | 36.97 | 35.55 | 37.12 | 36.32 |
|  | DINOv2 | 54.19 | 53.08 | 54.78 | 53.53 |
| GPT-4o | ResNet-18 | 38.33 | 37.01 | 38.41 | 36.06 |
|  | DINOv2 | 55.01 | 54.05 | 54.96 | 53.70 |

integrated with the two MLLMs. We believe this gap stems from the specific nature of our evaluation task, which is not open-ended medical reasoning but rather a constrained, visual instruction-following evaluation. While MedGemma possesses specialized medical knowledge, GPT-4o's massive scale and training on vastly diverse datasets have endowed it with powerful general visual reasoning and instruction-following capabilities.

Specifically, we hypothesize that GPT-4o's edge comes from its superior ability to parse the descriptive, often non-clinical language of the checklists (e.g., "bull's-eye lesions," "butterfly rash") and precisely map these concepts to visual features. In contrast, while MedGemma is fine-tuned on medical data, its smaller scale may slightly limit its raw visual-language alignment and nuanced instruction-following abilities. Most importantly, we see the comparable performance of MedGemma as a validation of our framework's flexibility, demonstrating that MAGIC is not dependent on a single, closed-source model and can be adapted using accessible alternatives.

# 6 Conclusion

In this work, we introduced MAGIC, a novel semi-automated framework designed to refine Diffusion Models by effectively integrating expert-enhanced clinical knowledge. Our approach uniquely leverages the visual reasoning capabilities of MLLMs to interpret and apply expert-defined checklists, thereby guiding DMs to produce images with high clinical fidelity while significantly reducing the burden on human experts. Our experiments demonstrate that MAGIC, substantially improves the clinical quality of synthesized skin disease images, as validated by both quantitative metrics like FID scores and qualitative assessments by dermatologists. Furthermore, augmenting training data with images generated by MAGIC led to significant enhancements in downstream classification accuracy for skin diseases, even in few-shot scenarios. These results underscore the efficacy of our AI-Expert collaboration paradigm in translating nuanced clinical criteria into actionable feedback for generative models. We acknowledge that our framework's performance is linked to the capabilities of the MLLM used. However, MAGIC is designed to be model-agnostic and flexible. This flexibility is also an advantage, as the framework's performance will naturally improve with the continually advancing interpretive capabilities of MLLMs. Beyond image synthesis, MAGIC demonstrates a task-centric alignment paradigm: instead of adapting MLLMs to niche medical tasks, it adapts tasks to the strengths of general-purpose MLLMs by decomposing domain knowledge into attribute-level checklists. This task-centric alignment is particularly valuable given that the most powerful MLLMs are often proprietary, and training domain-specific MLLMs is costly. This design offers a scalable and reliable path for leveraging foundation models in specialized domains.

## Acknowledgment

We thank the anonymous reviewers for their insightful comments and suggestions. This work was supported by the NSF EPSCoR-Louisiana Materials Design Alliance (LAMDA) program #OIA-1946231.

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

# Appendix

## A Additional Implementation Details

In this section, we present additional implementation details of our proposed method.

### A.1 Pre-Feedback Fine-tuning

For textual inversion, we learn the text embedding for each skin condition through various prompts. These prompts are used to ensure robust learning of the text embedding across different phrasings and contexts:

```
skin_disease_prompt = [
    "a photo of a {}",
    "a rendering of a {}",
    "a cropped photo of the {}",
    "the photo of a {}",
    "a close-up photo of a {}",
    "a cropped photo of a {}",
    "a photo of the {}",
    "a photo of one {}",
    "a close-up photo of the {}",
    "a rendition of the {}",
    "a rendition of a {}" ]
```

The text embeddings are learned don't the entire training set. The AdamW optimizer is used with a learning rate of $5 \times 10^{-4}$.

For LoRA, the rank $r$ is set to 32, and the learning rate is $5 \times 10^{-6}$ for AdamW optimizer.

### A.2 MLLM Score Processing for Preference Pairs

To translate 5-dimensional binary MLLM scores into preference signals for DPO, pairs of generated images are processed. For each image in a pair, its 5 binary scores are summed to get $S_1$ and $S_2$. If $\max(S_1, S_2) \leq 2$, both images are deemed low quality (outcome e.g., $[0, 0]$). If $\min(S_1, S_2) = 5$, or if $S_1 = S_2 > 2$, the pair is marked "both win" (e.g., $[1, 1]$). Otherwise, if $S_1 > S_2$, the first image is the "winner" (e.g., $[1, 0]$); if $S_2 > S_1$, the second wins (e.g., $[0, 1]$). This determines preferred/non-preferred samples for DPO loss computation. The distribution of these outcomes is in Table 6.

### A.3 DPO fine-tuning

We conduct DPO fine-tuning for 128 iterations and for each iteration, 8 pairs (16 images) will be sampled. The denoise strength $\gamma$ is set to 0.3. The DPO loss will be computed with the feedback. We utilize AdamW optimizer with a learning rate of 0.0001.

### A.4 Classifier Training

We utilize the Adam optimizer with a learning rate of 0.01 and a step learning rate scheduler that reduces the learning rate to 0.1 of its previous value every 50 epochs. The classifier is trained for 200 epochs to ensure stable results. Each result reported in the table represents the average of five runs with different random seeds.

## B Expert Designed Checklist

We enclose the checklist we used in the experiment in this section. For each skin condition, we design 5 checklist evaluations from the perspective of `[Location, Lesion Type, Shape/Size, Color, Texture]` to capture the visual concept from the synthetic data. The details are shown in Table 14.

## C Automate Evaluation via MLLMs

For each pair of data, we use the following prompt to collect feedback from ChatGPT-4o:

```
prompt = f'''Evaluate images against the
```

```
following checklist:
{condition_checklist}
Return a list indicating whether
it satisfies each checklist
item (1 for satisfied, 0 otherwise).
Only the list of results should
be returned. Expected format:
[1, 0, 1, 0, 0]'''
```

# D   Addtional Results

## D.1   Distribution of Feedback

For each pair of data, our approach categorizes feedback into three types: both win ($[w = 0, w = 1]$), both lose ($[l = 0, l = 1]$), and one better than the other ($[w = 0, l = 1]$ or $[l = 0, w = 1]$). We present the distribution of feedback received during DPO training in Table 6.

## D.2   More examples of image pairs

We provide two more image pairs in Fig. 7

## D.3   Results on SCIN and PAD-UFES-20

The SCIN dataset [62], collected via a voluntary image donation platform from Google Search users in the United States, typically includes up to three images per case, each evaluated by up to three dermatologists. This diagnostic process yields a weighted skin condition label for each case. To ensure label accuracy for our study, we selected the condition with the highest weight as the definitive label, discarding ambiguous cases where multiple conditions had equal probabilities. Our analysis concentrated on the 10 most prevalent classes in the real world. Given that the SCIN dataset exhibits an imbalanced class distribution, we first sampled a uniformly distributed test set, following methodologies similar to ImageNet-LT [38]. Furthermore, guided by approaches like that of [48], we employed our MAGIC-DPO framework to generate additional synthetic images for each condition, aiming to augment the test set towards a more uniform distribution. Further details on the dataset distribution are provided in Table 8. However, experiments conducted with this augmented SCIN dataset yielded suboptimal results, potentially attributable to inherent noise within the dataset, a challenge noted in works such as [27].

Our MAGIC framework's effectiveness is further validated on the SCIN dataset, with detailed performance for both ResNet18 and DINOv2 classifiers presented in Table 9 and Table 10. For the ResNet18 classifier on SCIN, models trained on real data achieved an accuracy of 23.13%. Standard T2I augmentation slightly decreased this to 22.60% ($-0.5\%$), while I2I augmentation offered a modest improvement to 24.13% ($+1.0\%$). In contrast, our MAGIC framework demonstrated more substantial gains: MAGIC-RFT increased accuracy to 26.58% ($+3.5\%$), and MAGIC-DPO further improved it to 29.43% ($+6.3\%$). A similar trend was observed with the DINOv2 classifier, which had a baseline accuracy of 30.61% on real SCIN data. T2I augmentation reduced accuracy to 28.18% ($-2.4\%$), and I2I provided a small increase to 32.15% ($+1.5\%$). Both MAGIC strategies again outperformed these: MAGIC-RFT achieved 33.82% accuracy ($+3.2\%$), while MAGIC-DPO led with 35.65% ($+5.0\%$). These results on the SCIN dataset consistently show the advantages of leveraging MAGIC, with both RFT and DPO components enhancing performance over standard augmentation techniques, and DPO often yielding the highest accuracy.

To quickly evaluate the cross-dataset generalizability of our method, we identified four overlapping classes between the hospital-grade PAD-UFES-20 dataset and Fitzpatrick17k subset (ACK, BCC, MEL, and SCC), and ran the MAGIC-DPO pipeline. The results, presented in the Table 11, demonstrate that the MAGIC framework is generalizable to hospital-grade datasets.

## D.4   Score change during training

Figure 6 illustrates how the clinical quality of generated images, assessed by the number of satisfied expert-defined criteria, evolves throughout the feedback-guided training phase of our MAGIC framework. Initially, images from the Pre-trained model and the fine-tuned Text-to-Image (T2I) model satisfy very few criteria, with average scores of 0.3 and 0.5, respectively. Even the fine-tuned Image-to-Image (I2I) model, at the beginning of feedback training (Iteration 0), achieves an average of only 1.4 criteria met. As the model receives more feedback and training progresses (Iterations 32 through 128), a significant improvement is observed. The distribution of scores progressively shifts towards satisfying a higher number of clinical criteria, with the average number of criteria met increasing steadily from 1.4 to 3.0 by Iteration 128. This trend clearly demonstrates the diffusion

model's ability to learn from and adapt to the expert-derived feedback over time, resulting in generated images that are increasingly more aligned with clinical requirements for medical accuracy.

## D.5    Evaluation with Other Backbones

A stronger backbone such as DINOv2 starts with a clear advantage: it has a complex transformer architecture and is pre-trained on a massive dataset, enabling it to extract robust image features from the start. In contrast, a smaller model like ResNet-18 struggles with the limited and challenging real data. The differing gains (+9.02% for ResNet18 vs. +5.12% for DINOv2) therefore highlight a key finding: our augmentation framework provides the most significant benefit in data-scarce or model-constrained scenarios.

However, the fact that MAGIC still substantially boosts the performance of a powerful model like DINOv2 is a strong testament to the quality of our synthetic data. It demonstrates that MAGIC generates images with clinically accurate features that even a strong classifier cannot extract from the limited real dataset alone. This conclusion is supported by our few-shot experiments in Table 3. When the training set was reduced to just 10% (310 images), MAGIC-DPO provided a sizable +10.94% accuracy improvement for the DINOv2 classifier. This shows that as data becomes more scarce, the value of our synthetic augmentation becomes even more pronounced. To further explore this interesting effect, we have run additional experiments with other classifier backbones of various sizes. The results, presented in the Table 12, are consistent with our observations.

## D.6    Effect of Checklist Granularity

As shown in Table 13, this analysis leads to two key insights: (1) A detailed, structured checklist is critical for the framework's success, as shown by the significant performance jump from the "Coarse" to the "Structured" checklist. (2) There may be diminishing returns after a certain level of detail is achieved, as shown by the smaller gain when moving from the 5-criteria to the 9-criteria checklist. This suggests our original 5-criteria checklist was already capturing the most essential features for high-quality generation.

## D.7    Discussion about Diversity

There are two distinct and important aspects of diversity:

- Inter-Site Diversity: Can the model generate the same condition with clinically appropriate, site-specific features (e.g., does Lupus on the scalp look different from Lupus on the face)?

- Intra-Site Diversity: Can the model generate multiple, varied appearances of the same condition at the same site (e.g., 100 different-looking examples of "Lupus on the face")?

The MAGIC framework is designed to address both of these challenges, as elaborated below.

**1. Inter-Site Diversity**: The model's ability to render a condition with site-specific features is driven by the synergy between our I2I pipeline and the structured checklists. The I2I pipeline grounds the generation process in a specific anatomical context by starting with a real source image (e.g., a scalp with hair). Guided by the feedback loop, the model is then tasked with generating features that satisfy the clinical checklist within the visual and anatomical constraints of that source image. While many of the selected conditions do not exhibit strong site-specificity, our expert-designed checklists are nuanced enough to include these manifestations wherever applicable. For example, the checklist for "Lupus Erythematosus" explicitly guides the model toward a "symmetric butterfly rash across the cheeks" when the target is the face, and a "discoid or coin-shaped lesion" otherwise. Over time, the MLLM-driven feedback rewards the model for plausibly blending the target lesion features with the source anatomical context.

**2. Intra-Site Diversity**: As noted in our initial rebuttal, our framework enhances diversity primarily through variation in the source images. To generate 100 diverse images of "Lupus Erythematosus on the face," we begin with 100 different real source images of faces, which naturally contain diversity in skin tone, age, gender, and so on. Our I2I process transforms the lesion on each unique face into lupus while preserving the source's individual characteristics. The resulting synthetic images are therefore as diverse as the original source images. Furthermore, this potential is not limited to the labeled training set, as our framework can effectively use unlabeled images as a source for generation, dramatically expanding the pool of available image contexts. Additionally, the model generates diverse outputs even when starting from the exact same source image. The denoising diffusion process is inherently stochastic, beginning with a randomly noised vector and denoising it into a different final image. Our DPO fine-tuning ensures that these random variations remain within the manifold of what is clinically plausible, resulting in meaningful variations in lesion presentations.

Here, we qualitatively confirm that images generated by our MAGIC framework exhibit both types of diversity, as shown by Fig. 8

# E    Discussions

While our MAGIC framework demonstrates significant promise, several exciting avenues for future work could enhance its efficiency and adaptability in clinical settings. The current approach relies on LoRA for efficient fine-tuning, but exploring alternative Parameter-Efficient Fine-Tuning (PEFT) methods, such as Visual Prompt Tuning (VPT) [29, 64, 56, 67, 61, 36], could offer different trade-offs in performance and computational cost, especially when adapting to new visual concepts. Furthermore, our few-shot experiments highlight the framework's potential to leverage unlabeled data, which could be formalized into a robust semi-supervised learning paradigm to further mitigate data scarcity. In a real-world scenario, diagnostic models must evolve as new disease data becomes available. Integrating principles from continual learning and online adaptation [71, 39, 70, 13] would enable the generative model to learn new skin conditions incrementally without suffering from catastrophic forgetting of previously learned ones. Additionally, we will investigate our pipeline's potential in real-world dermatological applications, such as synthesizing data for skin neglected tropical diseases (skin NTDs). This group of skin conditions impacts millions of people but suffers from serious data scarcity [59]. Future work could also explore the broader utility and privacy implications of such synthetic data replacement strategies [10].

# F    Limitations

The efficacy of our MAGIC framework, like similar feedback-driven approaches, is naturally guided by the detail within the expert-crafted checklists and the continually advancing interpretive capabilities of Multimodal Large Language Models (MLLMs). The scope of conditions and populations within the dermatology datasets utilized (Fitzpatrick17k and SCIN) provides the foundation for the current findings, and extending this work to even broader and more varied datasets presents an exciting avenue for future research. While MAGIC demonstrates considerable potential in dermatology, its promising AI-Expert collaboration paradigm also invites future exploration and adaptation to enhance synthetic data generation in other medical imaging fields, each with its unique visual characteristics and clinical requirements.

# G    Failure Analysis of GPT-4o

Based on a qualitative review of the MLLM's evaluations, we identify several recurring challenges and potential failure modes, which are detailed below. A systematic analysis of these failure modes is a valuable direction for future research:

- Subtle Textural and Morphological Details: The MLLM can struggle with very fine surface textures or lesions that are not well-defined. For example, small papules or pustules can be difficult for the model to assess, especially when their color is indistinguishable from the surrounding skin.

- Complex Color Nuances: Differentiating between similar shades (e.g., pink vs. reddish) or accurately interpreting colors on darker skin tones, such as the "purple/dark brown with grayish scales" described for Psoriasis, can be difficult from a 2D image alone.

- Inferring 3D Characteristics: Features that imply three-dimensionality, such as the "firm nodules" of Prurigo Nodularis or the "pearly bump" of Basal Cell Carcinoma, are inherently challenging to assess from a single 2D photograph.

- Ambiguity and Confounding Factors: For example, in one case, GPT-4o incorrectly identified medicine powder on a patient's skin as a white, hypopigmented scale. Without the accompanying clinical record, a human dermatologist would likely find it difficult to distinguish this from the image alone. This highlights that for both humans and AI, visual data can be ambiguous, and other factors like suboptimal lighting or the lack of clinical metadata can impede a purely visual assessment.

# H    Ethical Considerations and Safeguards

Our MAGIC framework is designed with privacy in mind. This "factorized transformation" preserves only the high-level anatomical context while overwriting the fine-grained lesion details. This dissociates the original identity from the new condition, which both enhances privacy and reduces the risk of the classifier learning spurious correlations. While this approach is designed to be privacy-conscious, we acknowledge that it does not offer the formal guarantees of methods like Differential Privacy. Additionally, privacy risks can be further minimized by running open-source or HIPAA-compliant MLLMs locally as evaluators. While a formal, quantitative analysis guaranteeing the complete removal of all identifying features was beyond the scope of this work, we agree that aligning diffusion models while removing identifying cues is an important future direction for medical and other privacy-critical domains.

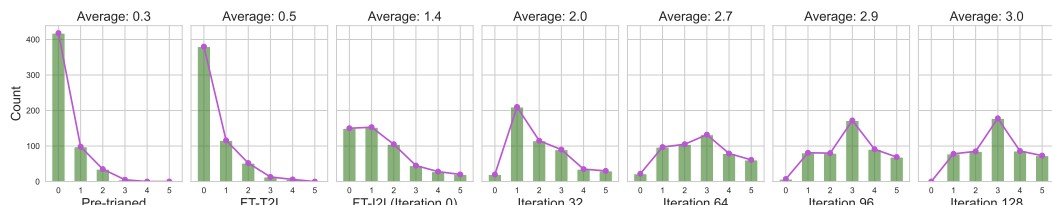

Figure 6: Feedback distribution as training progresses.

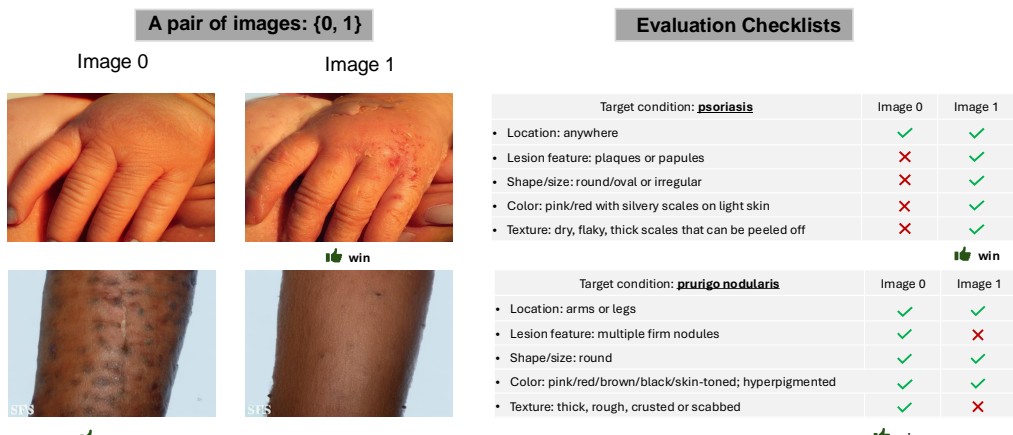

Figure 7: Two image pairs with the corresponding checklist.

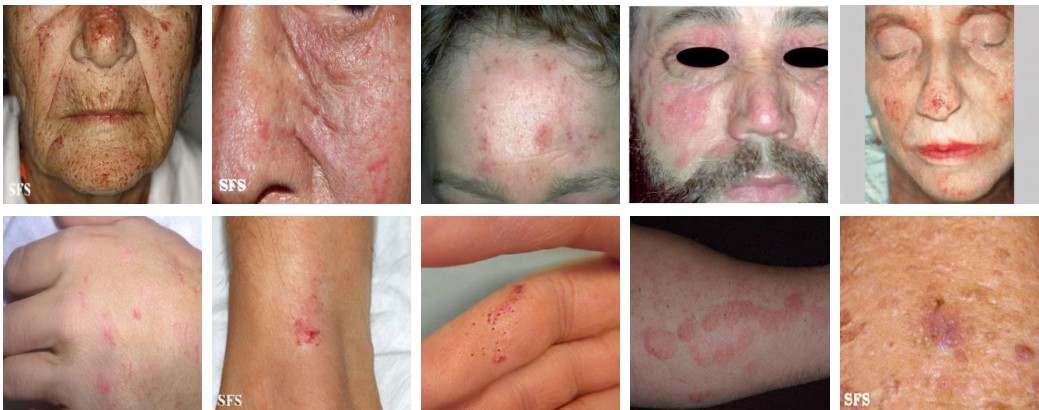

Figure 8: First row: diverse images for "Lupus Erythematosus on the face". Second row: diverse images of "Lupus Erythematosus" generated on different anatomical sites.

Table 6: Distribution of feedback

|  | **both win** | **only one win** | **both lose** |
|---|---|---|---|
| **count** | 295 | 397 | 332 |

Table 7: Skin Condition Distribution for Fitzpatrick17k

| **Skin Condition** | **Real Training** | **Real Test** | **Synthetic** |
|---|---|---|---|
| Acne | 92 | 91 | 93 |
| Actinic Keratosis | 88 | 87 | 164 |
| Allergic Contact Dermatitis | 215 | 215 | 181 |
| Basal Cell Carcinoma | 234 | 234 | 154 |
| Eczema | 102 | 102 | 166 |
| Erythema Multiforme | 118 | 118 | 155 |
| Folliculitis | 171 | 171 | 114 |
| Granuloma Annulare | 106 | 105 | 148 |
| Keloid | 78 | 78 | 135 |
| Lichen Planus | 246 | 245 | 151 |
| Lupus Erythematosus | 205 | 205 | 172 |
| Melanoma | 130 | 131 | 155 |
| Mycosis Fungoides | 91 | 91 | 165 |
| Pityriasis Rosea | 96 | 97 | 156 |
| Prurigo Nodularis | 85 | 85 | 152 |
| Psoriasis | 326 | 327 | 165 |
| Sarcoidosis | 174 | 175 | 162 |
| Scabies | 170 | 169 | 176 |
| Squamous Cell Carcinoma | 290 | 291 | 175 |
| Vitiligo | 83 | 83 | 161 |
| Total | 3100 | 3100 | 3100 |

Table 8: Skin Condition Distribution for SCIN

| **Skin Condition** | **Real Training** | **Real Test** | **Synthetic** |
|---|---|---|---|
| Eczema | 409 | 36 | 0 |
| Urticaria | 178 | 34 | 0 |
| Folliculitis | 104 | 35 | 33 |
| Tinea | 72 | 34 | 58 |
| Psoriasis | 57 | 39 | 70 |
| Herpes Simplex | 49 | 36 | 76 |
| Acne | 44 | 31 | 80 |
| Herpes Zoster | 41 | 29 | 82 |
| Pityriasis rosea | 41 | 32 | 82 |
| Tinea Versicolor | 27 | 34 | 93 |
| Total | 1022 | 340 | 574 |

Table 9: Performance of ResNet18-based classifiers trained on real and synthetic data for SCIN.

| Training data | Acc | F1 | Prec | Rec |
|---|---|---|---|---|
| Real | 23.13 | 10.94 | 12.20 | 10.70 |
| + T2I | 22.60 | 10.44 | 12.43 | 10.96 |
|  | -0.5 | -0.5 | +0.2 | +0.3 |
| + I2I | 24.13 | 10.90 | 12.03 | 11.06 |
|  | +1.0 | 0.0 | -0.2 | +0.4 |
| **+ MAGIC** | 26.58 | 11.69 | 15.79 | 11.89 |
| RFT | +3.5 | +0.7 | +3.6 | +1.2 |
| **+ MAGIC** | **29.43** | **12.16** | **18.18** | **11.47** |
| DPO | **+6.3** | **+1.2** | **+6.0** | **+0.8** |

Table 10: Performance of DINOv2-based classifiers trained on real and synthetic data for SCIN.

| Training data | Acc | F1 | Prec | Rec |
|---|---|---|---|---|
| Real | 30.61 | 18.37 | 21.23 | 17.45 |
| + T2I | 28.18 | 17.48 | 20.23 | 16.15 |
|  | -2.4 | -0.9 | -1.0 | -1.3 |
| + I2I | 32.15 | 20.10 | 23.80 | 19.06 |
|  | +1.5 | +1.7 | +2.6 | +1.6 |
| **+ MAGIC** | 33.82 | 20.08 | 24.16 | 18.70 |
| RFT | +3.2 | +1.7 | +2.9 | +1.2 |
| **+ MAGIC** | **35.65** | **21.39** | **24.00** | **19.40** |
| DPO | **+5.0** | **+3.0** | **+2.8** | **+1.9** |

Table 11: Performance (%) of MAGIC-DPO on PAD-UFES-20.

| Model | Setting | Accuracy (%) | F1 (%) | Precision (%) | Recall (%) |
|---|---|---|---|---|---|
| ResNet-18 | Real | 63.02 | 51.97 | 61.07 | 48.34 |
| | MAGIC | 70.65 | 58.48 | 62.40 | 51.65 |
| DINOv2 | Real | 67.88 | 60.50 | 66.29 | 57.10 |
| | MAGIC | 73.85 | 63.81 | 67.17 | 59.41 |

Table 12: Classifier accuracy (%) on real data vs. MAGIC-augmented training. Gain denotes absolute improvement.

| Model | #Params (M) | Pre-train Dataset | Real Acc (%) | MAGIC Acc (%) | Gain (%) |
|---|---|---|---|---|---|
| ResNet-18 | 12 | ImageNet-1k | 29.31 | 38.33 | +9.02 |
| ResNet-50 | 26 | ImageNet-1k | 35.56 | 46.24 | +10.68 |
| ViT-B/16 | 86 | ImageNet-21k | 45.22 | 51.44 | +6.22 |
| DINOv2 | 87 | LVD-142M | 49.89 | 55.01 | +5.12 |

Table 13: Effect of checklist granularity on performance (%).

| Model | Setting | Acc (%) | F1 (%) | Prec (%) | Rec (%) |
|---|---|---|---|---|---|
| ResNet-18 | w/ Coarse Checklist (1 sentence) | 32.83 | 30.58 | 29.75 | 31.18 |
| | w/ Fine-grained Checklist (7 criteria) | 38.33 | 37.01 | 38.41 | 36.06 |
| | w/ Highly Fine-grained Checklist (9 criteria) | 39.39 | 37.86 | 40.64 | 39.75 |
| DINO-v2 | w/ Coarse Checklist (1 sentence) | 51.16 | 52.66 | 52.17 | 52.69 |
| | w/ Fine-grained Checklist (7 criteria) | 55.01 | 54.05 | 54.96 | 53.70 |
| | w/ Highly Fine-grained Checklist (9 criteria) | 55.88 | 55.60 | 56.56 | 55.51 |

Table 14: Skin Conditions and Their Checklist Properties

| Skin Condition | Checklist Details |
|---|---|
| Acne | 1. Location: Face, forehead, chest, shoulders, upper back ( areas with many oil glands)
2. Lesion Type: Bumps including comedones (whiteheads, blackheads) and inflamed pimples (papules, pustules, nodules)
3. Shape/Size: Small clogged-pore bumps; larger tender nodules/cysts in severe cases
4. Color: Red or skin-colored bumps (may appear purple/brown on dark skin); blackheads have dark plug, whiteheads have white tip
5. Texture: Oily or shiny skin with multiple bumps; some lesions with pus or crust if ruptured |
| Actinic keratosis | 1. Location: Sun-exposed areas (face, scalp, ears, neck, forearms, backs of hands)
2. Lesion Type: Rough, scaly patch or small crusty bump
3. Shape/Size: Flat or slightly raised lesion, usually under 2.5 cm
4. Color: Pink, red, or brownish, possibly with a yellowish crust; on darker skin can appear gray or dark
5. Texture: Dry, coarse, sandpaper-like surface; may have a hard or wart-like feel |
| Allergic contact dermatitis | 1. Location: Where allergen contacts skin (hands, face, eyelids, neck, etc.)
2. Lesion Type: Red patches often with small blisters (vesicles) or swelling
3. Shape/Size: Irregular shape following exposure pattern; size depends on contact area
4. Color: Pink to red on light skin; can be darker, purple, or brownish on dark skin
5. Texture: May be weepy, crusty, or scaly; inflamed and swollen in acute cases |
| Basal cell carcinoma | 1. Location: Sun-exposed areas (face, nose, ears, neck, scalp, shoulders)
2. Lesion Type: Pearly or waxy bump/nodule, or flat scaly patch with a raised edge
3. Shape/Size: Small, round/oval; can ulcerate or develop a central depression
4. Color: Translucent or pearly on fair skin; brown/black or glossy dark on darker skin
5. Texture: Smooth, shiny surface; can crust or scab with central ulceration |

| Skin Condition | Checklist Details |
| --- | --- |
| Eczema | 1. Location: Flexural areas (inner elbows, behind knees), hands, ankles, neck, eyelids, cheeks
2. Lesion Type: Patches or plaques, sometimes with small blisters or bumps
3. Shape/Size: Ill-defined patches varying in size; often bilateral or symmetric
4. Color: Red or pink on lighter skin; purple, gray, or dark brown on darker skin
5. Texture: Dry, flaky, or scaly; can become thick and leathery (lichenification) |
| Erythema multiforme | 1. Location: Hands, feet, arms, legs, can involve mucous membranes (lips, mouth, eyes)
2. Lesion Type: Target (bull's-eye) lesions with concentric rings
3. Shape/Size: Round lesions (1-3 cm) with a dark center, pale ring, and outer red ring
4. Color: Center is dark red/purple, ring is lighter or pink, outer zone is red; on dark skin, may be grayish or hyperpigmented center
5. Texture: Mostly flat but can have a blistered or raised center |
| Folliculitis | 1. Location: Hair-bearing areas prone to friction or shaving (beard, scalp, underarms, legs, buttocks)
2. Lesion Type: Small pustules or red papules centered around hair follicles
3. Shape/Size: Clusters of 2-5 mm bumps; each with a central hair
4. Color: Red or pink on light skin; darker or hyperpigmented on dark skin; pus may appear white/yellow
5. Texture: Dome-shaped, often with a fluid-filled top; can crust if ruptured |
| Granuloma annulare | 1. Location: Hands, feet, wrists, ankles (localized); can appear on trunk/limbs if generalized
2. Lesion Type: Smooth, firm bumps (papules) forming rings; typically non-scaly
3. Shape/Size: Annular (ring-shaped) up to a few cm wide; papules are a few mm each
4. Color: Skin-colored, pink, or reddish; can appear purple on darker skin
5. Texture: Generally smooth; little to no flaking or crust |

| Skin Condition | Checklist Details |
| --- | --- |
| Keloid | 1. Location: Scars on chest, shoulders, earlobes, jawline, or any site of skin injury
2. Lesion Type: Overgrown scar tissue extending beyond the original wound
3. Shape/Size: Raised, irregularly shaped scar; can be small or grow large over time
4. Color: Pink or red on lighter skin; darker, purple or brown on darker skin
5. Texture: Smooth, hairless, firm/rubbery; shiny surface |
| Lichen planus | 1. Location: Wrists, forearms, ankles, scalp, nails, mouth, genitals
2. Lesion Type: Flat-topped papules; can form plaques or lines from scratching
3. Shape/Size: Polygonal, 2-10 mm papules
4. Color: Violaceous (purple) on light skin; gray-brown or hyperpigmented on dark skin
5. Texture: Shiny surface with fine white lines (Wickham's striae); can be scaly if scratched |
| Lupus erythematosus | 1. Location: Face (butterfly rash across cheeks/nose); can appear on scalp/ears; photosensitive areas
2. Lesion Type: Flat or slightly raised rash (malar/butterfly); discoid lesions can be scaly and scarred
3. Shape/Size: Butterfly rash covers the bridge of nose and both cheeks; discoid lesions are coin-shaped (1-3 cm)
4. Color: Pink-red on light skin; can be darker red or hyperpigmented on darker skin
5. Texture: Malar rash smooth or slightly raised; discoid can be rough/scaly with scarring |
| Melanoma | 1. Location: Can appear anywhere (trunk, limbs, face, nails); in darker skin, often on palms/soles or under nails
2. Lesion Type: Atypical mole or patch; irregular shape and color
3. Shape/Size: Asymmetric, often >6 mm, with notched/bumpy edges
4. Color: Multiple shades (brown, black, red, white, blue); on dark skin, often very dark with variation
5. Texture: Smooth early; may become raised, crusted, or ulcerated if advanced |

| Skin Condition | Checklist Details |
| --- | --- |
| Mycosis fungoides | 1. Location: Usually non-sun-exposed areas (buttocks, lower abdomen, thighs); can spread more widely later
2. Lesion Type: Patches (like eczema), plaques (thickened), or tumor nodules (advanced)
3. Shape/Size: Irregular shapes, patches often a few cm wide; plaques larger/thicker; nodules can be several cm
4. Color: Pink-red to reddish-brown; darker or hyperpigmented on darker skin
5. Texture: Dry, scaly for patches; plaques thicker/scaly; nodules can be smooth or ulcerated |
| Pityriasis rosea | 1. Location: Trunk (back, chest, abdomen) primarily; occasionally upper arms, thighs
2. Lesion Type: Herald patch (large oval) followed by multiple smaller oval patches/papules
3. Shape/Size: Herald patch ~2-6 cm; daughter lesions ~1-2 cm; often align in 'Christmas tree' pattern
4. Color: Pink/salmon on light skin; gray, brown, or purplish on dark skin
5. Texture: Fine collarette scale at inner edge; not typically thick or crusty |
| Prurigo nodularis | 1. Location: Arms, legs, upper back, shoulders, scalp, areas easily reached for scratching
2. Lesion Type: Firm, itchy nodules, often with a crusted or scabbed top
3. Shape/Size: Round nodules 1-3 cm; multiple lesions often present
4. Color: May be pink, red, brown, black, or skin-toned; older lesions can be hyperpigmented
5. Texture: Thick, rough; scabs from scratching; firm to touch |
| Psoriasis | 1. Location: Elbows, knees, scalp, lower back; can affect nails, palms, soles, or be widespread
2. Lesion Type: Well-demarcated plaques with thick, scaly surface; can also be smaller papules
3. Shape/Size: Round/oval or irregular plaques; can range from small patches to large areas
4. Color: On light skin, pink/red with silvery scales; on dark skin, purple/dark brown with grayish scales
5. Texture: Dry, flaky scales that can be peeled off; underlying skin may bleed (Auspitz sign) |

| Skin Condition | Checklist Details |
|---|---|
| Sarcoidosis | 1. Location: Face (nose, cheeks – lupus pernio), shins ( erythema nodosum), scars/tattoos, can be widespread
2. Lesion Type: Firm plaques, nodules, or discolored patches; red bumps on shins in erythema nodosum
3. Shape/Size: Plaques are broad and raised; nodules can be 1–5 cm; patchy discolorations vary
4. Color: Purplish or red-brown lumps; can be lighter/darker patches on dark skin; scars can turn red
5. Texture: Smooth, firm or rubbery; some lesions (erythema nodosum) are tender lumps under the skin |
| Scabies | 1. Location: Finger webs, wrists, waist, buttocks, genitals, armpits; in infants: palms, soles, scalp
2. Lesion Type: Tiny burrows (thin, wavy lines) plus small itchy bumps or vesicles
3. Shape/Size: Burrows ~5–15 mm long; bumps ~1–2 mm in clusters
4. Color: Skin-toned to pink/red; on darker skin, may appear darker or hyperpigmented
5. Texture: Scratch marks, crusted spots from itching; burrows feel like slight ridges |
| Squamous cell carcinoma | 1. Location: Sun-exposed areas (face, ears, lips, hands), chronic scars, or wounds; can appear on mucosal surfaces
2. Lesion Type: Crusty or scaly bump, ulcer, or plaque; can have raised borders or a central depression
3. Shape/Size: Firm nodule or patch, >1 cm if untreated; may grow rapidly
4. Color: Pink/red on lighter skin; brown or darker on brown/ Black skin; can show white/yellow keratin
5. Texture: Rough, thick, crusted surface; may bleed or ulcerate; firm on palpation |
| Vitiligo | 1. Location: Face (around eyes, mouth), hands, feet, arms, legs , genitals; can occur anywhere on body
2. Lesion Type: Depigmented patches with well-defined borders; hair may turn white in affected area
3. Shape/Size: Irregular shapes; can start small and enlarge over time, often symmetrical
4. Color: Completely white or pale compared to surrounding skin ; high contrast on darker skin
5. Texture: Normal skin texture (no scaling or thickening), only color is lost |

