# OpenReview forum: "Doctor Approved: Generating Medically Accurate Skin Disease Images through AI-Expert Feedback"
_NeurIPS.cc/2025/Conference — NeurIPS 2025 poster_

### Official Review · Reviewer_WddE · 2025-06-21

**Clarity:** 4
**Significance:** 3
**Originality:** 3
**Rating:** 5
**Confidence:** 4

**Summary:**

The authors introduce a novel diffusion-based framework for generating skin images, titled "Medically Accurate Generation of Images through AI-Expert Collaboration." This method effectively converts criteria set by experts into practical feedback for synthesizing images of Diffusion Models, markedly enhancing clinical precision and decreasing the workload on human operators.  Within this framework, human experts are mainly needed to: (1) create checklists based on credible sources that are easily verifiable by an MLLM, and (2) monitor the MLLM’s feedback on synthetic images throughout the training of T2I DMs. Experiments show that their approach significantly enhances the clinical quality of synthesized images of skin diseases, aligning closely with dermatologist evaluations. Furthermore, incorporating these synthesized images into the training data boosts diagnostic accuracy by 9.02% on a complex skin disease classification task involving 20 conditions, and by 13.89% in a few-shot scenario.

**Questions:**

Results are limited to one single modality. Do the authors think the method can be adapted to the modalities with ease? Can the authors comment on the key parts to be changed for adoption?

Can the authors illustrate whether the model is able to generate enough diverse data for pictures of similar anatomic sites?

At line 185, the authors mentioned that "only synthetic images are sent to GPT-4o API services and no real patient images are processed by the MLLM, to preserve privacy." However, the clinical images that are the subject of this paper are large field-of-view images, which can contain PHI. As the proposed framework only alters the image with respect to the target disease category, the privacy concerning certain portions of the image will most likely remain intact. In this case, even the synthetic image may still contain private content, and the sharing of data remains a concern. Nonetheless, this issue does not affect the validity of the paper's results, as it can be addressed by using a HIPAA-compliant, local model in the future.

In Figure 2: the direction of the arrow between the classes on the left side of the image does not align with the text description in the caption.

**Ethical Concerns:**

["NO or VERY MINOR ethics concerns only"]

**Final Justification:**

The authors replied all my comments and offered to update the camera ready paper with addition details provided in their comments

**Limitations:**

Results are only limited to one single modality; further testing involving different modalities may be required
Diversity in the generation is a question. The authors did not illustrate if the model is able to generate enough diverse data for pictures of similar sites.

**Quality:**

4

**Strengths And Weaknesses:**

**Strengths**
-  The need for augmented data in training medical models is a significant problem. More importantly, the development of methods that are grounded in concepts and are related to the lesion appearance and diagnostic properties is an essential need in the field. This work touches this issue and provides a solution.
- The claims presented in the article are backed with sufficient techinical justification. In this respect the authors used a combination of image generation quality metrics (e.g. FID) and also some diagnostically related metrics (the expert checklist) to gauge the quality of the generated images.
- The use of MLLM during the evaluation of the generated images increases the efficiency of the approach as it decreases the burden on the clinician/expert.
- The evaluation results from the AI-Human collaborative system are fed back to the diffusion model to fintune the model. The authors used 2 alternative ways to do this (Reward-model guided fine tuning and Direct Preference Optimization) and validated their performance through a classification experiment.
- The paper is well organized and clearly written. It provides adequate information to the reader to understand the approach
- Numerous diffusion-based data generation models have been discussed in the literature. The authors build on the literature by incorporating human feedback through "skin condition checklists" to enhance the grounding of generation models.

**Weakness**
- The authors used a generic AI model to check the generated images against the skin condition checklist generated by experts. It is not clear how capable this model is of assessing the images in terms of the queried conditions.
- It is not clear if the authors did any qualitative evaluation conducted by an expert. In the conclusions section they mentioned about qualitative evaluations but it is not clear if they are refering to "skin condition checklist" evaluation by the MLLM or some manual evaluation.

---

> ### Author Rebuttal · Authors · 2025-07-31
>
> We thank the reviwer for the recognition of our method's technical justification and clear presentation, and we appreciate the constructive feedback provided.
>
> ## W1: Reliability of GPT-4o
> ---
> We address this in two parts: by detailing our framework's built-in safeguards and by presenting two new empirical analyses.
>
> 1. **Framework Design to Mitigate Hallucination:** The MAGIC framework is explicitly designed to minimize the risk of MLLM hallucination through our AI-Expert collaboration paradigm. The MLLM is not asked to perform open-ended reasoning. Instead, its role is constrained to evaluating an image against a predefined clinical checklist. These checklists, designed by dermatologists, decompose complex medical concepts into simple, visually verifiable features. For instance, while an MLLM may not intrinsically understand "Lupus Erythematosus," it can effectively verify "symmetric butterfly rash across the cheeks". This approach transforms a complex diagnostic reasoning task into a series of closed-question evaluations, which are far less susceptible to unconstrained hallucination.
>
>     Even so, we agree that an analysis of GPT-4o's performance is important. To that end, we conducted a new study where we had GPT-4o evaluate 100 real images from an internal dataset with clinical records confirmed by in-person visits. The MLLM was tasked with describing each image based on the five criteria (location, lesion type, shape/size, color, and texture). We then invited a board-certified dermatologist to assess these descriptions. As shown in the table below, the dermatologist assigned an **alignment score** of 3 or greater (on a 5-point scale) to approximately **86%** of the image-description pairs. Notably, many of the lower-scoring examples were also identified by the dermatologist as being visually ambiguous and challenging for a human to assess from an image alone.
>
> |Score(↑)|1|2|3|4|5|
> |-|-|-|-|-|-|
> |# Examples|8|6|16|21|49|
>
> 2. **Choice of MLLM and Comparative Analysis:** The MAGIC framework is designed to be model-agnostic, allowing for the flexibility to use the most suitable MLLM for a given task. For this foundational study, we selected GPT-4o for its exceptional visual reasoning capabilities. However, we agree that a comparative analysis using open-source models is a valuable addition. To that end, we have conducted new experiments testing our MAGIC framework with an open-source MLLM, Google DeepMind's **MedGemma-4B**[1], an open model for medical text and image comprehension. We used MedGemma as an evaluator to assess the same set of 1024 pairs of synthesized images that GPT-4o had assessed. We observed that MedGemma is comparable to GPT-4o as an evaluator in aligning the DM using the MAGIC framework. This demonstrates the flexibility of our MAGIC framework, which is adaptive to both large, closed-source, genric models and smaller, open-source, domain-specific alternatives. The results are shown in the table below.
>
> |MLLM|Classifier|Accuracy(%)|F1(%)|Precision(%)|Recall(%)|
> |-|-|-|-|-|-|
> |GPT-4o|ResNet-18|38.33|37.01|38.41|36.06|
> |GPT-4o|DINOv2|55.01|54.05|54.96|53.70|
> |MedGemma|ResNet-18|36.97|35.55|37.12|36.32|
> |MedGemma|DINOv2|54.19|53.08|54.78|53.53|
>
> [1]. MedGemma Technical Report. (2025)
>
> We'll include the results and analyses in the future version of the work.
>
> ## W2: Expert Evaluation
> ---
> A separate qualitative evaluation was conducted entirely by human medical experts. This is detailed in **Section 5.1** under "Expert evaluation on generated images". In this study, we engaged medical experts to evaluate the synthetic data against the five clinical criteria. The results of this manual evaluation are summarized in **Fig.4(d)**.
>
> Additionally, we also conducted a blinded Turing test to assess the clinical realism of our generated images. We created a mixed set of 500 real images and 500 synthetic images generated by MAGIC-DPO. An expert dermatologist, who was not involved in the project, was asked to rate the likelihood of each of the 1,000 images being synthetic on a 5-point scale given the condition label. The expert's ratings for the real and synthetic images followed a remarkably similar distribution, and the expert tended not to make absolute predictions. This indicates that the expert could not confidently distinguish the synthetic images from the real clinical photos. This experiment complements the quality evaluation in Fig. 4(d) by directly assessing realism.
>
> ||1-very likely|2-likely|3-neutral|4-unlikely|5-very unlikely|
> |-|-|-|-|-|-|
> |Real|9|151|90|249|1|
> |Syn|11|131|83|267|8|
>
> ## Q1, L1: Adapting MAGIC to other Modalities
> ---
> This is an excellent question. While our current work focuses on dermatological photos, the core MAGIC framework is designed to be modality-agnostic. Its fundamental principle is to integrate structured, domain-specific knowledge into the generative process via expert checklists that are executable by MLLMs. As we note in our conclusion, the framework is adaptable to other medical fields, each with its "unique visual characteristics and clinical requirements". For example, adapting the method to modalities like radiology or digital pathology would be feasible and would primarily require changes to the following three components:
>
> - **The Expert Checklists**: This is the most critical and domain-specific component. For a new modality, relevant experts (e.g., radiologists) would need to create new checklists that capture the essential visual features of pathologies in that domain. For a chest X-ray, this could include criteria for lesion location, opacity, borders, and size.
> - **The Base Diffusion Model**: Before applying the MAGIC feedback loop, the base diffusion model would need to be fine-tuned on a representative dataset from the new modality. This preliminary step is crucial for teaching the model the fundamental anatomy and characteristics of the new image type.
> - **The MLLM Evaluator**: While powerful MLLMs are increasingly generalist, optimal performance might be achieved by selecting a model best suited for the target modality. For instance, one might choose an MLLM specifically fine-tuned on radiological images and reports to ensure the most accurate interpretation of the expert checklist.
>
> ## Q2, L2: Generate Diverse Data
> ---
> The ability to generate diverse data is a key strength of our framework, which we achieve through its novel I2I mechanism and can verify both qualitatively and quantitatively.
> 1. **Leveraging Source Image Diversity via the I2I Pipeline**: The primary driver of diversity is our use of the I2I pipeline. Our process transforms a real source image to a new target condition while preserving the original anatomical context. This means the diversity of the generated images—in terms of skin tone, age, gender, lighting, and viewing angle—is directly inherited from the diversity of the real source images. For example, to generate 100 diverse images of "lupus on the face," we can start with 100 diverse real-world images of faces (with any other condition), and the resulting synthetic images will retain this rich variation. And this is where MAGIC shines. As demonstrated in our few-shot learning experiments, the framework is not limited to using labeled images from the training set as a source. We can use any relevant image, including unlabeled data, to generate new examples. This dramatically expands the potential for diversity, as the framework's capacity to generate varied data grows with the number of external source images one can access.
> 2. **Qualitative and Quantitative Evidence**: We can see the model's ability to generate diverse conditions on the same anatomy in **Fig.2**. By comparing the first and last columns, it is clear that the framework can successfully transform a single image into a new target condition with distinct features, while preserving the underlying anatomical information. This increases dataset diversity by allowing different conditions to be realistically rendered in various contexts. We will add more qualitative comparisons between real and synthesized images in our work. Quantitatively, the improvement in classification performance also indicates the improved generalizability resulting from enhanced diversity. However, we agree that rigorous quantification of diversity to assess the quality of synthesized images is an important question, and we will investigate this in our future work.
>
> ## Q3: Privacy Concern about GPT-4o
> ---
> The MAGIC framework transforms the lesion and dissociates the condition from any particular person, reducing the risk of privacy leakage. Additionally, privacy risks can be further minimized by running open-source, HIPAA-compliant (as suggested by the reviewer) MLLMs locally as evaluators. Moreover, synthesizing images that preserve a higher level of differential privacy would require dedicated DP strategies. While a formal, quantitative analysis guaranteeing the complete removal of all identifying features was beyond the scope of this work, we agree that aligning diffusion models while removing identifying cues is an important future direction for medical and other privacy-critical domains. We will add a note on this to our discussion in the "**Ethical Considerations and Safeguards**" section of the final paper.
>
> ## Q4: Image-caption Mismatch in Fig.2
> ---
> We thank the reviewer for their careful reading and for catching this error. The caption text is accurate, and we will correct the arrows in the figure for the final version of the paper.

---

> > ### Comment · Reviewer_WddE · 2025-08-05
> >
> > I would like to thank the authors for replying to the comment reviews. I think I was not clear in my question about diversity.
> >
> > Yes, Figure 2 shows different example lesions/conditions generated for the same site. But my question was whether the model can generate different disease/condition appearances on different backgrounds (e.g., sites, etc.). For example, a particular condition (e.g., Sarcoidosis or Erythema Multiforme as in Fig 2) does not display the same features at every site, so does the model adapt to the change in visual appearance of the condition for different sites? Similarly, if one wants to generate the same condition at the same site, can they get diverse images (unlike the latent walk type of presentation in Figure 2) or very similar images?

---

> ### Author Response · Authors · 2025-08-06
> **Response to the Follow-up Question about Diversity**
>
> Thank you for the follow-up and for clarifying your question about diversity!
>
> ### Question Formulation
> These questions highlight two distinct and important aspects of diversity:
>
> - **Inter-Site Diversity:** Can the model generate the same condition with clinically appropriate, site-specific features (e.g., does Lupus on the scalp look different from Lupus on the face)?
> - **Intra-Site Diversity:** Can the model generate multiple, varied appearances of the same condition at the same site (e.g., 100 different-looking examples of "Lupus on the face")?
> ---
> ### Response
> The MAGIC framework is designed to address both of these challenges, as elaborated below.
>
> 1. **Inter-Site Diversity:** The model’s ability to render a condition with site-specific features is driven by the synergy between our **I2I pipeline** and the **structured checklists**. The I2I pipeline grounds the generation process in a specific anatomical context by starting with a real source image (e.g., a scalp with hair). Guided by the feedback loop, the model is then tasked with generating features that satisfy the clinical checklist within the visual and anatomical constraints of that source image. While many of the selected conditions do not exhibit strong site-specificity, our expert-designed checklists are nuanced enough to include these manifestations wherever applicable. For example, the checklist for "Lupus Erythematosus" explicitly guides the model toward a "symmetric butterfly rash across the cheeks" when the target is the face, and a "discoid or coin-shaped lesion" otherwise. Over time, the MLLM-driven feedback rewards the model for plausibly blending the target lesion features with the source anatomical context.
>
> 2. **Intra-Site Diversity:** As noted in our initial rebuttal, our framework enhances diversity primarily through variation in the **source images**. To generate 100 diverse images of "Lupus Erythematosus on the face," we begin with 100 different real source images of faces, which naturally contain diversity in skin tone, age, gender, and so on. Our I2I process transforms the lesion on each unique face into lupus while preserving the source's individual characteristics. The resulting synthetic images are therefore as diverse as the original source images. Furthermore, this potential is not limited to the labeled training set, as our framework can effectively use unlabeled images as a source for generation, dramatically expanding the pool of available image contexts.
>
>     Additionally, the model generates diverse outputs even when **starting from the exact same source image**. The denoising diffusion process is inherently stochastic, beginning with a randomly noised vector and denoising it into a different final image. Our DPO fine-tuning ensures that these random variations remain within the manifold of what is clinically plausible, resulting in meaningful variations in lesion presentations.
> ---
> ### Commitment for Final Paper
> In response to this question, we have qualitatively confirmed that images generated by our MAGIC framework exhibit both types of diversity. Unfortunately, due to policy, we cannot provide images in this response that directly demonstrate these forms of diversity. To illustrate these capabilities in the final manuscript, we will add two figures:
> - **For Inter-Site Diversity:** A figure showing a single condition (e.g., Lupus Erythematosus) generated on different anatomical sites (e.g., the face and scalp) to demonstrate the model's adaptive rendering.
> - **For Intra-Site Diversity:** A figure displaying multiple, diverse images for a single condition and site (e.g., "Lupus on different faces"). This will include examples generated from different source images to show contextual variation, as well as from the same source image to highlight stochastic variation.
> ---
> ### Concluding Summary
> In summary, our framework achieves two critical forms of diversity. Inter-site diversity is learned by combining the anatomical context from source images with site-specific clinical checklists. Intra-site diversity is driven both by using a wide variety of source images and by the inherent stochastic nature of the diffusion process itself. We commit to adding new figures to the final manuscript to visually substantiate both of these capabilities.
>
> We again thank you for this insightful question, as it helps us to further clarify and strengthen our work. Please let us know if this response has addressed all of your concerns. We would be happy to provide any further clarification needed.

---

> > ### Comment · Reviewer_WddE · 2025-08-06
> >
> > Thank you for the further clarification and details. My score reflects that I believe what you describe is the case, but the details were missing from the paper. Thank you for offering to add those details to the paper.
> > I have no additional comments.

---

> > > ### Author Response · Authors · 2025-08-06
> > > **Thank you for your time and constructive feedback!**
> > >
> > > Thank you for your valuable guidance throughout the review process. We sincerely appreciate your belief in our work. Your insightful questions have been instrumental in helping us clarify and strengthen the manuscript for its final version. We will be active until the end of the discussion period and welcome any further questions you may have. Thank you!

---

### Official Review · Reviewer_Embq · 2025-06-27

**Clarity:** 3
**Significance:** 4
**Originality:** 4
**Rating:** 5
**Confidence:** 4

**Summary:**

This paper proposes MAGIC, a novel framework integrating AI-generated expert feedback via MLLMs into diffusion models for the generation of clinically accurate synthetic dermatological images.

The method notably reduces human annotation workloads and demonstrates significant performance improvements in downstream classification tasks.

**Questions:**

x. How sensitive is MAGIC's performance to inaccuracies or omissions in expert-provided checklists? Could the authors elaborate on robustness tests regarding potential checklist variability?

x. What insights can the authors share about the performance drop when exceeding a certain ratio of synthetic data (p > 0.5)? Could there be practical strategies to maximize benefits from synthetic images while avoiding potential domain shifts?

x. Can the authors provide insights on the failure modes or common inaccuracies observed when GPT-4o evaluates dermatological images? Are there specific lesion characteristics that are particularly challenging for the MLLM to assess correctly?

**Ethical Concerns:**

["NO or VERY MINOR ethics concerns only"]

**Final Justification:**

A timely, well-formatted paper for the healthcare applications. Recommend acceptance.

**Limitations:**

The paper accurately identifies reliance on high-quality expert-generated checklists as a potential limitation.

Further exploration into automating or systematically validating checklist creation could mitigate human bias.

**Quality:**

3

**Strengths And Weaknesses:**

Strengths:

+ Effectively leverages Multimodal Large Language Models for image evaluation, significantly reducing human annotation labor.

+ Demonstrates tangible performance improvements: a notable increase in diagnostic accuracy (+9.02% in general settings and +13.89% in few-shot scenarios).

+ Robust baseline comparisons (T2I, I2I) and well-justified use of Direct Preference Optimization (DPO) and Reward-based Fine-Tuning (RFT).


Maojr weakness:

-  The effectiveness of MAGIC heavily relies on checklist quality and clarity (highlighted in ablation studies on page 9). If dermatologists overlook key characteristics or checklist detail is insufficient, the method's performance may degrade significantly.

- While leveraging GPT-4o is innovative, the practical implications and reliability of AI-based medical judgments in clinical workflows are not discussed in depth.

---

> ### Author Rebuttal · Authors · 2025-07-31
>
> We thank the reviewer for their positive feedback and for recognizing our method's tangible impact on diagnostic accuracy, its innovative use of MLLMs, and the robustness of our experimental design. We also appreciate the constructive feedback.
> ## W1: Reliance on Checklist Quality
> ---
> This is an insightful question. Our ablation study was designed to demonstrate this very point: the quality of the expert-provided checklist is pivotal for aligning diffusion models with clinical reality. While the goal is to significantly reduce the expert labor burden by automating the repetitive evaluation of thousands of images, we believe that the initial, high-leverage task of ensuring checklist quality is an essential and manageable role for human experts.
>
> To ensure this quality in our work, we employed a robust, multi-step process.
> **First**: the checklists were crafted using multiple credible clinical sources for comprehensiveness.
> **Second**: these checklists were then cross-validated by several dermatologists to ensure clinical accuracy and minimize the risk of overlooking key diagnostic features.
>
> The results of our ablation study in **Table 4** confirm that this expert effort is critical; the detailed, well-structured checklist enabled a much larger accuracy boost compared to a coarse one. This underscores a core message of our paper: generating high-quality synthetic medical data requires successfully integrating granular expert knowledge into the generation process. While this represents a dependency on expert input, it also highlights the framework's primary strength: its ability to effectively translate nuanced clinical criteria into actionable feedback, guiding the diffusion models toward clinical reality.
>
> ## W2: Reliability of GPT-4o
> ---
> We address this in two parts: by detailing our framework's built-in safeguards and by presenting two new empirical analyses.
> 1. **Framework Design to Mitigate Hallucination**: The MAGIC framework is explicitly designed to minimize the risk of MLLM hallucination through our AI-Expert collaboration paradigm. The MLLM is not asked to perform open-ended reasoning. Instead, its role is constrained to evaluating an image against a predefined clinical checklist. These checklists, designed by dermatologists, decompose complex medical concepts into simple, visually verifiable features. For instance, while an MLLM may not intrinsically understand "Lupus Erythematosus," it can effectively verify "symmetric butterfly rash across the cheeks". This approach transforms a complex diagnostic reasoning task into a series of closed-question evaluations, which are far less susceptible to unconstrained hallucination.
>
>     Even so, we agree that an analysis of GPT-4o's performance is important. To that end, we conducted a new study where we had GPT-4o evaluate 100 real images from an internal dataset with clinical records confirmed by in-person visits. The MLLM was tasked with describing each image based on the five criteria (location, lesion type, shape/size, color, and texture). We then invited a board-certified dermatologist to assess these descriptions. As shown in the table below, the dermatologist assigned an **alignment score of 3** or greater (on a 5-point scale) to approximately **86%** of the image-description pairs. Notably, many of the lower-scoring examples were also identified by the dermatologist as being visually ambiguous and challenging for a human to assess from an image alone.
>
> |Score(↑)|1|2|3|4|5|
> |-|-|-|-|-|-|
> |# Examples|8|6|16|21|49|
>
> 2. **Choice of MLLM and Comparative Analysis:** The MAGIC framework is designed to be model-agnostic, allowing for the flexibility to use the most suitable MLLM for a given task. For this foundational study, we selected GPT-4o for its exceptional visual reasoning capabilities. However, we agree that a comparative analysis using open-source models is a valuable addition. To that end, we have conducted new experiments testing our MAGIC framework with an open-source MLLM, Google DeepMind's **MedGemma-4B**[1], an open model for medical text and image comprehension. We used MedGemma as an evaluator to assess the same set of 1024 pairs of synthesized images that GPT-4o had assessed. We observed that MedGemma is comparable to GPT-4o as an evaluator in aligning the DM using the MAGIC framework. This demonstrates the flexibility of our MAGIC framework, which is adaptive to both large, closed-source, genric models and smaller, open-source, domain-specific alternatives. The results are shown in the table below.
>
> |MLLM|Classifier|Accuracy(%)|F1(%)|Precision(%)|Recall(%)|
> |-|-|-|-|-|-|
> |GPT-4o|ResNet-18|38.33|37.01|38.41|36.06|
> |GPT-4o|DINOv2|55.01|54.05|54.96|53.70|
> |MedGemma|ResNet-18|36.97|35.55|37.12|36.32|
> |MedGemma|DINOv2|54.19|53.08|54.78|53.53|
>
>
>
> We'll include the results and analyses in the future version of the work. Additionally, we agree that the direct use of AI for medical judgments is a complex topic involving crucial discussions on ethics, regulation, and reliability. A full exploration of these issues is beyond the scope of this foundational technical paper, but our work contributes by improving the underlying data quality, which is a prerequisite for any reliable clinical AI system.
>
> [1]. MedGemma Technical Report. (2025)
>
> ## Q1: MAGIC's Sensitivity/Robustness to Checklist Quality
> ---
> Our work presents a direct robustness test for this, and we have conducted an additional experiment to explore this sensitivity further.
>
> First, the ablation study in **Section 5.2, "Effect of Checklist Quality**," serves as our primary robustness test. By comparing the performance of models trained with data generated using "Coarse" vs. "Structured" checklists, we explicitly tested the system's sensitivity to checklist detail. The results in **Table 4** show that performance is indeed highly sensitive to checklist quality, confirming that granular expert knowledge is a critical driver for generating high-quality synthetic data.
>
> To explore the upper bounds of this effect, we conducted a new experiment. We had a dermatologist craft even more fine-grained checklists with nine criteria (Location, Distribution, Lesion Type, Shape/Size, Border, Elevation, Texture, Color, and Translucency/Content). We then used these 9-criteria checklists in our MAGIC framework. We observed a small, additional improvement over our original 5-criteria structured checklist, as shown in the table below.
>
> |ResNet-18|Accuracy(%)|F1(%)|Precision(%)|Reall(%)|
> |-|-|-|-|-|
> |w/CoarseChecklist (1sentence)|32.83|30.58|29.75|31.18|
> |w/Fine-grainedChecklist (7-criteria)|38.33|37.01|38.41|36.06|
> |w/HighlyFine-grainedChecklist (9-criteria)|39.39|37.86|40.64|39.75|
>
> |DINO-v2|Accuracy(%)|F1(%)|Precision(%)|Reall(%)|
> |-|-|-|-|-|
> |w/CoarseChecklist (1sentence)|51.16|52.66|52.17|52.69|
> |w/Fine-grainedChecklist (7-criteria)|55.01|54.05|54.96|53.70|
> |w/HighlyFine-grainedChecklist (9-criteria)|55.88|55.60|56.56|55.51|
>
> This combined analysis leads to two key insights: **(1)** A detailed, structured checklist is critical for the framework's success, as shown by the significant performance jump from the "Coarse" to the "Structured" checklist. **(2)** There may be diminishing returns after a certain level of detail is achieved, as shown by the smaller gain when moving from the 5-criteria to the 9-criteria checklist. This suggests our original 5-criteria checklist was already capturing the most essential features for high-quality generation.
>
> ## Q2: Maximize Benefits from Synthetic Images
> ---
> Our insight is that when the proportion of synthetic data becomes too high (ρ > 0.5), the classifier begins to overfit to the specific nuances and distribution of the generated images, thereby losing some of its ability to generalize to the real-world test data. Even high-quality synthetic data has a slightly different distribution from real clinical photos. The most effective practical strategy to mitigate this, and the one employed in our work, is to use a mix of real and synthetic data in each training batch.
>
> ## Q3: Failure Analysis of GPT-4o
> ---
> Based on a qualitative review of the MLLM's evaluations, we identified several recurring challenges and potential failure modes, which are detailed below. A systematic analysis of these failure modes is a valuable direction for future research.
>
> - Subtle Textural and Morphological Details: The MLLM can struggle with very fine surface textures or lesions that are not well-defined. For example, small papules or pustules can be difficult for the model to assess, especially when their color is indistinguishable from the surrounding skin.
> - Complex Color Nuances: Differentiating between similar shades (e.g., pink vs. reddish) or accurately interpreting colors on darker skin tones, such as the "purple/dark brown with grayish scales" described for Psoriasis, can be difficult from a 2D image alone.
> - Inferring 3D Characteristics: Features that imply three-dimensionality, such as the "firm nodules" of Prurigo Nodularis or the "pearly bump" of Basal Cell Carcinoma, are inherently challenging to assess from a single 2D photograph.
> - Ambiguity and Confounding Factors: For example, in one case, GPT-4o incorrectly identified medicine powder on a patient's skin as a white, hypopigmented scale. Without the accompanying clinical record, a human dermatologist would likely find it difficult to distinguish this from the image alone. This highlights that for both humans and AI, visual data can be ambiguous, and other factors like suboptimal lighting or the lack of clinical metadata can impede a purely visual assessment.
>
> ### Limitations
> ---
> We thank the reviewer for this excellent suggestion. We agree that exploring methods to automate or more systematically validate the checklist creation process is critical. This is a valuable avenue for future research that builds directly on our framework and we will add this to our discussion of future work.

---

### Official Review · Reviewer_Jzjk · 2025-07-01

**Clarity:** 3
**Significance:** 2
**Originality:** 2
**Rating:** 5
**Confidence:** 3

**Summary:**

This paper introduces MAGIC (Medically Accurate Generation of Images through AI–Expert Collaboration), a novel semi-automated framework to generate clinically accurate dermatological images using diffusion models (DMs), and multimodal large language models (MLLMs) like GPT-4o. MAGIC fine-tunes image generation via expert-designed criteria translated into machine-readable feedback. It can leverage two learning paradigms: Reward-model Fine-Tuning (RFT) or Direct Preference Optimization (DPO), with DPO yielding superior results.

**Questions:**

1. I wonder how did you make sure, GPT4o is reasonable to be used as an expert?
2. For expert evaluation in your experiment part, did you invite real medical doctors to evaluate the generated images?
3. Section 3.2 first sentence need citations to make sure its correctness.
4. Also you didn't start from a random noise, I wonder what the benefit of your method is. Do you have any experiments to show it?

**Ethical Concerns:**

["NO or VERY MINOR ethics concerns only"]

**Final Justification:**

The reviewer has tried their best to solve my problems, and the follow-up experiments successfully showed the benefits of their proposed methods. Based on the re-justification, i raise my score to 5 accordingly.

**Limitations:**

1. Technical dependence: Relies on proprietary MLLM capabilities that may change over time
2. Domain specificity: Requires adaptation of checklists and evaluation criteria for each new medical domain

**Quality:**

3

**Strengths And Weaknesses:**

Strenghts: 1. The method is sound to me. This paper introduces a practical and semi-automated way to incorporate human feedback using MLLMs, reducing costly manual evaluations.
2. The problems they proposed are critical, especially that the generated images look pretty fake in the medical domain.
3. The experiment results demonstrate the method's superior performance. The ablation studies are sufficient, and experimental details are great.

Weaknesses: 1. Only two datasets are used for the experiment.
2. The authors simply use GPT4-o for the assessment. I doubt that if GPT4-o is 100% percent correct to serve as an expert.
3. Missing comparisons with other medical-specific synthetic data generation approaches.

---

> ### Author Rebuttal · Authors · 2025-07-31
>
> We thank the reviewer for the thoughtful assessment and valuable feedback. We are very pleased that they found our method sound, the problem critical, and our experiments comprehensive and well-supported.
>
> ## W1: Choice of Datasets
> ---
> From the limited number of available clinical datasets, we strategically chose Fitzpatrick17k and SCIN as they represent real-world challenges, such as complex variations and data scarcity. To further evaluate the generalizability of our method, we also performed a new evaluation on a new dataset: PAD-UFES-20. To quickly evaluate our method, we identified four overlapping classes between the PAD-UFES-20 dataset and our Fitzpatrick17k subset (ACK, BCC, MEL, and SCC), and ran the MAGIC-DPO pipeline. The results, presented in the tables below, demonstrate that the MAGIC framework is also generalizable to hospital-grade datasets.
>
> |ResNet-18|Accuracy(%)|F1(%)|Precision(%)|Recall(%)|
> |-|-|-|-|-|
> |Real|63.02|51.97|61.07|48.34|
> |MAGIC|70.65|58.48|62.40|51.65|
>
> |DINOv2|Accuracy(%)|F1(%)|Precision(%)|Recall(%)|
> |-|-|-|-|-|
> |Real|67.88|60.50|66.29|57.10|
> |MAGIC|73.85|63.81|67.17|59.41|
>
> ## W2 & Q1: Reliability of GPT-4o
> ---
> We address this in two parts:
> 1. **Framework Design to Ensure Reasonableness:** We do not use the MLLM as an autonomous expert. Instead, its role is carefully constrained within our AI-Expert collaboration paradigm. The MLLM is not asked to perform open-ended reasoning; its role is limited to evaluating an image against a predefined clinical checklist. These checklists, designed by dermatologists, decompose complex medical concepts into simple, visually verifiable features. For instance, while an MLLM may not intrinsically understand "Lupus Erythematosus," it can effectively verify a feature like a "symmetric butterfly rash across the cheeks". This approach transforms a complex diagnostic reasoning task into a series of closed-question evaluations, making the MLLM's role both reasonable and verifiable.
>
>     Even so, we agree that an analysis of GPT-4o's performance is important. To that end, we conducted a new study where we had GPT-4o evaluate 100 real images from an internal dataset with clinical records confirmed by in-person visits. The MLLM was tasked with describing each image based on the five criteria (location, lesion type, shape/size, color, and texture). We then invited a board-certified dermatologist to assess these descriptions. As shown in the table below, the dermatologist assigned an **alignment score** of 3 or greater (on a 5-point scale) to approximately **86%** of the image-description pairs. Notably, many of the lower-scoring examples were also identified by the dermatologist as being visually ambiguous and challenging for a human to assess from an image alone.
>
> |Score(↑)|1|2|3|4|5|
> |-|-|-|-|-|-|
> |# Examples|8|6|16|21|49|
>
> 2. **Choice of MLLM and Comparative Analysis:** The MAGIC framework is designed to be model-agnostic, allowing for the flexibility to use the most suitable MLLM for a given task. For this foundational study, we selected GPT-4o for its exceptional visual reasoning capabilities. However, we agree that a comparative analysis using open-source models is a valuable addition. To that end, we have conducted new experiments testing our MAGIC framework with an open-source MLLM, Google DeepMind's **MedGemma-4B**[1], an open model for medical text and image comprehension. We used MedGemma as an evaluator to assess the same set of 1024 pairs of synthesized images that GPT-4o had assessed. We observed that MedGemma is comparable to GPT-4o as an evaluator in aligning the DM using the MAGIC framework. This demonstrates the flexibility of our MAGIC framework, which is adaptive to both large, closed-source, genric models and smaller, open-source, domain-specific alternatives. The results are shown in the table below.
>
> |MLLM|Classifier|Accuracy(%)|F1(%)|Precision(%)|Recall(%)|
> |-|-|-|-|-|-|
> |GPT-4o|ResNet-18|38.33|37.01|38.41|36.06|
> |GPT-4o|DINOv2|55.01|54.05|54.96|53.70|
> |MedGemma|ResNet-18|36.97|35.55|37.12|36.32|
> |MedGemma|DINOv2|54.19|53.08|54.78|53.53|
>
> [1]. MedGemma Technical Report. (2025)
>
> We'll include the results and analyses in the future version of the work.
>
> ## W3: Comparision with Other Medical Synthesis Method
> ---
> In our 'Related Works' section, we discuss several relevant studies in dermatological data synthesis. Our experimental comparisons were designed to be relevant to our core contribution, which is the expert-feedback alignment loop. Our work is directly compared against two SOTA methods: (i) **The method from [50]:** This represents an effective approach for skin disease image synthesis *without* an active expert-feedback loop. Our "+I2I" baseline is based on this work. (2) **The method from [46]:** This is a key attempt at aligning a medical image generator with expert feedback by training a reward function. Our MAGIC-RFT model is our adaptation and comparison to this methodology, tailored from a class-conditional DM to our T2I DM. Furthermore, by comparing with [50], we are also indirectly assessed against a method that has been systematically evaluated against other recognized approaches in the field, including [2, 39, 40]. This allows us to situate our performance within the broader context of existing literature. ([2, 39, 40, 46, 50] in our paper).
>
> ## Q2: Expert Evaluation on the Generated Images
> ---
> For the final assessment of the generated images, we have engaged **real medical experts** to evaluate the synthetic data based on our specific checklist criteria, and the results are present in **Fig.4 (d)**. Additionally, we also conducted a blinded Turing test to assess the clinical realism of our generated images. We randomly sampled a mixed set of 500 real images and 500 synthetic images generated by MAGIC-DPO. We also resized the images to the same dimensions. An expert dermatologist, who was not involved in the project, was asked to rate the likelihood of each of the 1,000 images being synthetic on a 5-point scale given the condition label. The expert's ratings for the real and synthetic images followed a remarkably similar distribution, and the expert tended not to make absolute predictions. This indicates that the expert could not confidently distinguish the synthetic images from the real clinical photos. This experiment complements the quality evaluation in Fig. 4(d) by directly assessing realism. We'll include this analysis in our work.
>
> ||1-very likely|2-likely|3-neutral|4-unlikely|5-very unlikely|
> |-|-|-|-|-|-|
> |Real|9|151|90|249|1|
> |Syn|11|131|83|267|8|
>
> ## Q3: Claim in the First Sentence of Section 3.2
> ---
> The claim is well-supported by prior work **[50]**, which demonstrated that an off-the-shelf diffusion model lacks the necessary "domain-specific knowledge" for this task. They attribute this to the model being unfamiliar with specific skin lesion concepts and semantics. Furthermore, fine-tuning diffusion models has become a common and necessary practice for synthetic augmentation in this field, as exemplified by recent studies, including [2, 28, 39, 40, 46, 50]. Following the reviewer's suggestion, we will add references, including [50], to the end of this sentence to make the citation explicit and properly credit these foundational observations. ([2, 28, 39, 40, 46, 50] in our paper).
>
> ## Q4: Benefits of Denoising from Partially Noised Images
> ---
> The benefit of starting from a partially noised image instead of pure Gaussian noise is that it enables a "factored transformation," where we can change the skin condition while preserving the underlying anatomy. This approach has several key advantages:
> - It helps to ensure the model develops a disentangled understanding of skin lesion concepts separate from other irrelevant visual cues.
> - It reduces semantic distortion during generation.
> - It helps prevent the downstream classifier from learning spurious correlations (e.g., associating a specific disease with a specific body part or background cue).
> - It accelerates the image generation process.
>
> In our experiments, the **"T2I"** method generates images from pure Gaussian noise, while the **"I2I"** method starts from a partially noised real image. The results in **Table 1** and **Table 2** clearly show the superiority of the I2I approach, where the T2I augmentation is outperformed by its I2I counterpart. Given this strong experimental evidence, we chose the more effective I2I pipeline for our MAGIC framework.
>
> ## L1: Technical Dependence on MLLMs
> ---
> We acknowledge that our framework's performance is linked to the capabilities of the MLLM used. However, MAGIC is designed to be model-agnostic and flexible. This flexibility is also an advantage, as the framework's performance will naturally improve with the continually advancing interpretive capabilities of MLLMs. As demonstrated in our response to **W2**, our framework can incorporate open-source models (e.g., MedGemma).
>
> ## L2: Domain Specificity
> ---
> The reviewer is correct that adapting MAGIC to new medical domains requires new, expert-designed checklists. This is an intended feature of our design. The core purpose of MAGIC is to translate specific, nuanced clinical knowledge into actionable feedback for a generative model. Therefore, achieving accuracy in any new domain fundamentally requires integrating targeted expert knowledge. We see this adaptability as a strength, allowing for rigorous, domain-aware data synthesis in other areas with their own unique visual characteristics and clinical requirements.
>
> ---
>
> We once again thank the reviewer for their valuable time and constructive feedback. We hope our detailed responses and new experimental results have fully addressed the concerns raised and have better highlighted the significance of our contributions. We would be grateful if you would consider these clarifications in your final evaluation of our work and are happy to answer any further questions you may have during the discussion period.

---

> > ### Comment · Reviewer_Jzjk · 2025-08-01
> > **Additional Question**
> >
> > I thank the authors for their feedback. I can see they have tried their best to solve all the problems i mentioned.
> >
> > I have an additional question following your new experiment. It seems that MedGemma-4B was beaten by GPT4o. I wonder if this is because GPT4o has better visual reasoning capabilities? Do you have any insights on this result?
> >
> > I will consider raising my score after this.

---

> > > ### Author Response · Authors · 2025-08-01
> > > **Insights On the Performance of GPT-4o vs. MedGemma-4B**
> > >
> > > We thank the reviewer for this insightful follow-up question. We agree that the superior performance of GPT-4o is likely attributable to its strong visual reasoning capabilities. The insights we gained from the results are as follows:
> > >
> > > We believe this performance gap stems from the specific nature of our **evaluation task**, which is not open-ended medical reasoning but rather a constrained, visual instruction-following evaluation. While MedGemma possesses specialized medical knowledge, GPT-4o's massive scale and training on vastly diverse datasets have endowed it with powerful general **visual reasoning and instruction-following capabilities**[1, 2].
> > >
> > > Specifically, we hypothesize that GPT-4o's edge comes from its superior ability to parse the descriptive, often non-clinical language of the checklists (e.g., "bull's-eye lesions," "butterfly rash") and precisely map these concepts to visual features. In contrast, while MedGemma is fine-tuned on medical data, its smaller scale may slightly limit its raw visual-language alignment and nuanced instruction-following abilities.
> > >
> > > Most importantly, we see the comparable performance of MedGemma not as a weakness, but as a resounding validation of our framework's flexibility. It demonstrates that MAGIC is not dependent on a single, closed-source model and can be effectively adapted using accessible alternatives. This adaptability is a core strength of our approach, empowering researchers to leverage the best tool for their specific needs, whether it be a large generalist model or a targeted specialist one.
> > >
> > > We hope these insights clarify the implications of our new results.
> > >
> > >
> > > [1] GPT-4o System Card, 2024
> > >
> > > [2] MedGemma Technical Report, 2025

---

> > > > ### Comment · Reviewer_Jzjk · 2025-08-01
> > > > **Thanks for the follow up.**
> > > >
> > > > Thanks for your answers. I will raise my score to 5 accordingly.

---

> > > > > ### Author Response · Authors · 2025-08-01
> > > > > **Thank you for your time and effort!**
> > > > >
> > > > > We appreciate your professional and constructive feedback, which made our work more solid and clearer. We will be active until the end of the discussion period and welcome any further questions you may have. Thank you!

---

### Official Review · Reviewer_d8Vt · 2025-07-15

**Clarity:** 2
**Significance:** 2
**Originality:** 2
**Rating:** 5
**Confidence:** 3

**Summary:**

This paper introduces a framework called MAGIC. They propose to fine-tune diffusion models to produce clinically accurate synthetic skin-disease images. The main contribution is to translate dermatologist-crafted check-lists into machine-readable feedback: a multimodal LLM (GPT-4o) scores each generated image on five visual criteria, and the diffusion model is aligned with either reward-weighted likelihood or DPO. Experiments on 2 datasets show that MAGIC-DPO raises ResNet-18 accuracy from 29 → 38 % and DINOv2 from 50 → 55 %, while more than half of the synthetic images satisfy ≥3 clinical criteria.

**Questions:**

Please discuss the weaknesses and here are some further questions for your paper:

How does MAGIC perform on unseen hospital-grade datasets e.g., PAD-UFES-20 or in a blinded dermatologist Turing test?

The I2I pipeline denoises real patient images conditioned on a new disease label. Could residual patient identity or background cues leak into the synthetic set, violating privacy regulations? You should discuss this problem

Fine-tuning with 1 024 feedback pairs took 128 iterations; please report wall-clock time and GPU hours to guide practitioners.

**Ethical Concerns:**

["NO or VERY MINOR ethics concerns only"]

**Final Justification:**

My main concerns on this paper is resolved. I will increase my final rating to Accept.

**Limitations:**

No limitation.

However, as you are a medical ML research, please add a candid section on misuse (e.g., fake dermatology imagery for fraud), bias across skin tones, and safeguards for releasing tuned diffusion weights.

**Paper Formatting Concerns:**

The format is ok to us.

**Quality:**

2

**Strengths And Weaknesses:**

**Strengths**

- Compared to the other medical imaging paper I reviewed in NeurISP this year, this paper is well structured and algorithmic steps are enumerated, which is good.

- It highlights an inexpensive path to reduce expert burden by pairing MLLM judges with concise clinical check-lists.

- The workflow in experiment is clear, dermatology synthesis -> prove it in the downstream tasks.

**Weaknesses**

- This is an application-only paper. It builds directly on prior DPO-diffusion work [54] and MLLM-judge paradigms; we can't say low novelty but the innovation is largely domain transfer.

- No analysis of hallucinations for the MLLM-as-a-judge process. Why not use some open-source model to replace GPT-4o and explore the hallucinations and potential bias?

- When the model scale of Classifier is larger, the performance increment is also lower. You should explore this marginal effect.

---

> ### Author Rebuttal · Authors · 2025-07-30
>
> We thank the reviewer for recognizing the quality of our work and our success in translating the MAGIC framework into tangible downstream performance improvements. We also sincerely appreciate the constructive feedback on our submission.
> ## W1: Applicaiton-oriented & Novelty
> ---
> While our framework leverages established methods like DPO-diffusion and MLLM-as-a-judge, this is the first work as far as we know that proposes a framework of combining expert domain knowledge and MLLM-based evaluation into actionable feedback for synthesizing domain-accurate images using DMs. We show significantly improved clinical accuracy while reducing the direct human workload in the dermatology domain, where generating clinically accurate images from limited data is non-trivial due to high intra-class variation and inter-class similarity of conditions. However, the core contribution is not merely a domain transfer but a new, semi-automated, and effective paradigm for image synthesis, whose potential could be extended to real-world applications beyond dermatology and even medical domains.
> ## W2: Analysis of hallucination & Open-source MLLM
> ---
> We address this in two parts: by detailing our framework's built-in safeguards and by presenting two new empirical analyses.
>
> 1. **Framework Design to Mitigate Hallucination:** The MAGIC framework is explicitly designed to minimize the risk of MLLM hallucination through our AI-Expert collaboration paradigm. The MLLM is not asked to perform open-ended reasoning. Instead, its role is constrained to evaluating an image against a predefined clinical checklist. These checklists, designed by dermatologists, decompose complex medical concepts into simple, visually verifiable features. For instance, while an MLLM may not intrinsically understand "Lupus Erythematosus," it can effectively verify "symmetric butterfly rash across the cheeks". This approach transforms a complex diagnostic reasoning task into a series of closed-question evaluations, which are far less susceptible to unconstrained hallucination.
>
>     Even so, we agree that an analysis of GPT-4o's performance is important. To that end, we conducted a new study where we had GPT-4o evaluate 100 real images from an internal dataset with clinical records confirmed by in-person visits. The MLLM was tasked with describing each image based on the five criteria (location, lesion type, shape/size, color, and texture). We then invited a board-certified dermatologist to assess these descriptions. As shown in the table below, the dermatologist assigned an **alignment score** of 3 or greater (on a 5-point scale) to approximately 86% of the image-description pairs. Notably, many of the lower-scoring examples were also identified by the dermatologist as being visually ambiguous and challenging for a human to assess from an image alone.
>
> |Score(↑)|1|2|3|4|5|
> |-|-|-|-|-|-|
> |# Examples|8|6|16|21|49|
>
>
> 2. **Choice of MLLM and Comparative Analysis:** The MAGIC framework is designed to be model-agnostic, allowing for the flexibility to use the most suitable MLLM for a given task. For this foundational study, we selected GPT-4o for its exceptional visual reasoning capabilities. However, we agree with the reviewer that a comparative analysis using open-source models is a valuable addition. To that end, we have conducted new experiments testing our MAGIC framework with an open-source MLLM, Google DeepMind's **MedGemma-4B**[1], an open model for medical text and image comprehension. We used MedGemma as an evaluator to assess the same set of 1024 pairs of synthesized images that GPT-4o had assessed. We observed that MedGemma is comparable to GPT-4o as an evaluator in aligning the DM using the MAGIC framework. This demonstrates the flexibility of our MAGIC framework, which is adaptive to both large, closed-source, genric models and smaller, open-source, domain-specific alternatives. The results are shown in the table below.
>
> |MLLM|Classifier|Accuracy(%)|F1(%)|Precision(%)|Recall(%)|
> |-|-|-|-|-|-|
> |GPT-4o|ResNet-18|38.33|37.01|38.41|36.06|
> |GPT-4o|DINOv2|55.01|54.05|54.96|53.70|
> |MedGemma|ResNet-18|36.97|35.55|37.12|36.32|
> |MedGemma|DINOv2|54.19|53.08|54.78|53.53|
>
> [1]. MedGemma Technical Report. (2025)
>
> We'll include the results and analyses in the future version of the work.
>
> ## W3: Analysis of Classifier Backbones
> ---
> A stronger backbone such as DINOv2  starts with a clear advantage: it has a complex transformer architecture and is pre-trained on a massive dataset, enabling it to extract robust image features from the start. In contrast, a smaller model like ResNet-18  struggles with the limited and challenging real data. The differing gains (**+9.02%** for ResNet18 vs. **+5.12%** for DINOv2) therefore highlight a key finding: our augmentation framework provides the most significant benefit in data-scarce or model-constrained scenarios.
>
> However, the fact that MAGIC still substantially boosts the performance of a powerful model like DINOv2 is a strong testament to the quality of our synthetic data. It demonstrates that MAGIC generates images with clinically accurate features that even a strong classifier cannot extract from the limited real dataset alone. This conclusion is supported by our few-shot experiments in **Table 3**. When the training set was reduced to just 10% (310 images), MAGIC-DPO provided a sizable **+10.94%** accuracy improvement for the DINOv2 classifier. This shows that as data becomes more scarce, the value of our synthetic augmentation becomes even more pronounced. To further explore this interesting effect, we have run additional experiments with other classifier backbones of various sizes. The results, presented in the table below, are consistent with our observations. We'll include the analysis in our work.
>
> |Model|#Params(M)|Pre-train Dataset|Real Acc(%)|MAGIC Acc(%)|Gain(%)|
> |-|-|-|-|-|-|
> |ResNet-18|12|ImageNet-1k|29.31|38.33|+9.02|
> |ResNet-50|26|ImageNet-1k|35.56|46.24|+10.68|
> |ViT-B/16|86|ImageNet-21k|45.22|51.44|+6.22|
> |DINOv2|87|LVD-142M|49.89|55.01|+5.12|
>
> ## Q1: Generalizability and Clinical Realism
> ---
> To directly address these questions, we have conducted two new experiments to evaluate the generalizability and clinical realism of our MAGIC framework.
>
> 1. **Generalizability to Unseen Dataset:** To quickly evaluate the cross-dataset generalizability of our method, we identified four overlapping classes between the PAD-UFES-20 dataset and our Fitzpatrick17k subset (ACK, BCC, MEL, and SCC), and ran the MAGIC-DPO pipeline. The results, presented in the tables below, demonstrate that the MAGIC framework is generalizable to hospital-grade datasets.
>
> |ResNet-18|Accuracy(%)|F1(%)|Precision(%)|Recall(%)|
> |-|-|-|-|-|
> |Real|63.02|51.97|61.07|48.34|
> |MAGIC|70.65|58.48|62.40|51.65|
>
> |DINOv2|Accuracy(%)|F1(%)|Precision(%)|Recall(%)|
> |-|-|-|-|-|
> |Real|67.88|60.50|66.29|57.10|
> |MAGIC|73.85|63.81|67.17|59.41|
>
> 2. **Blinded Dermatologist Evaluation:** We conducted a blinded Turing test to assess the clinical realism of our generated images. We randomly sampled a mixed set of 500 real images and 500 synthetic images generated by MAGIC-DPO. We also resized the images to the same dimensions. An expert dermatologist, who was not involved in the project, was asked to rate the likelihood of each of the 1,000 images being synthetic on a 5-point scale given the condition label. The expert's ratings for the real and synthetic images followed a remarkably similar distribution, and the expert tended not to make absolute predictions. This indicates that the expert could not confidently distinguish the synthetic images from the real clinical photos. This experiment complements the quality evaluation in Fig. 4(d) by directly assessing realism.
>
> ||1-very likely|2-likely|3-neutral|4-unlikely|5-very unlikely|
> |-|-|-|-|-|-|
> |Real|9|151|90|249|1|
> |Syn|11|131|83|267|8|
>
> We'll include the results and analyses in the future version of the work.
>
> ## Q2: Privacy Concern about I2I Pipeline
> ---
> Our MAGIC framework is designed with privacy in mind. This "factorized transformation" preserves only the high-level anatomical context while overwriting the fine-grained lesion details. This dissociates the original identity from the new condition, which both enhances privacy and reduces the risk of the classifier learning spurious correlations.
> While this approach is designed to be privacy-conscious, we acknowledge that it does not offer the formal guarantees of methods like Differential Privacy. Additionally, privacy risks can be further minimized by running open-source or HIPAA-compliant MLLMs locally as evaluators. While a formal, quantitative analysis guaranteeing the complete removal of all identifying features was beyond the scope of this work, we agree that aligning diffusion models while removing identifying cues is an important future direction for medical and other privacy‑critical domains.
>
> ## Q3: Computation Analysis
> ---
> We have included the table for wall-clock time on a single NVIDIA RTX 6000 Ada here. We will add the table to the Appendix.
> |Method|Config|Wall-Clock Time(h)|
> |-|-|-|
> |Textual Inversion|500 steps/token, 20tokens|4.05|
> |LoRA|3000 steps|0.44|
> |RFT|2024 feedback|2.01|
> |DPO|2024 feedback|3.77|
>
> ## Limitation
> ---
> We agree completely that a candid discussion on the broader impacts of this method is essential. Following this feedback, we will add a new "Ethical Considerations and Safeguards" section to our paper to discuss these concerns.
>
> ---
> We once again thank the reviewer for their valuable time and constructive feedback. We hope our detailed responses and new experimental results have fully addressed the concerns raised and have better highlighted the significance of our contributions. We would be happy to answer any further questions you may have during the discussion period.

---

> > ### Comment · Reviewer_d8Vt · 2025-08-07
> >
> > Thank you to the authors for your detailed and thoughtful response. Happly to see you conduct blinded dermatologist evaluation, which is very useful. Since my main concerns on this paper is resolved. I will increase my final rating to Accept. Good luck!

---

> > > ### Author Response · Authors · 2025-08-07
> > > **We appreciate your acknowledgement and the re-evaluation**
> > >
> > > We are glad that our response has addressed your concerns. Your feedback has been invaluable in strengthening our paper, and we are sincerely grateful for your time and effort. Thank you!

---

> ### Author Response · Authors · 2025-08-06
> **Follow-up on Rebuttal**
>
> Dear Reviewer,
>
> Thank you again for your time and effort in reviewing our paper. As the discussion period is ending soon, we wanted to respectfully follow up to see if our rebuttal has addressed your concerns. We remain available to provide any further clarifications you might need.
>
> The Authors

---

### Note · Authors · 2025-08-12

We sincerely thank the reviewers for an engaging and constructive discussion, which allowed us to substantially enhance the paper with new experiments and insights. These key additions, which will be integrated into the final manuscript, include:

- **Expanded Dermatologist Evaluation:** We conducted a Turing-style evaluation, where an expert dermatologist could not confidently distinguish our MAGIC-generated images from real clinical photos, confirming the method's high clinical realism. We also analyzed the reliability of GPT-4o as an evaluator by having a dermatologist assess the alignment of GPT-4o's generated captions against images from a clinically verified dataset.
- **Demonstrated Framework Flexibility and Generalizability:** We validated our framework's model-agnostic design by successfully using an open-source MLLM (MedGemma). Furthermore, we demonstrated its generalizability on a new, hospital-grade dataset (PAD-UFES-20).
- **New Ablation Study on Checklist Granularity:** Our experiments revealed two key insights: (1) A detailed, structured checklist is critical for success, shown by the significant performance jump from "Coarse" to "Structured" checklists. (2) Diminishing returns may occur after a certain level of detail is achieved, suggesting our original 5-criteria checklist effectively captured the most essential features for high-quality generation.
- **Clarified Definition of Generation Diversity:** The discussion helped us formulate a much clearer, twofold definition of diversity (inter-site and intra-site). To visually substantiate this, we will add figures to the manuscript demonstrating that our framework enables both forms of diversity.

Beyond these improvements, the discussion has sharpened the framing of the paper's primary contribution: **a novel and flexible paradigm for human-AI collaborative alignment**. Instead of adapting powerful MLLMs to niche tasks, our framework adapts the task to the MLLM's strengths by decomposing domain knowledge into attribute-based checklists with general concepts. This "task-centric" alignment is particularly valuable given that the most powerful MLLMs are often proprietary, and training domain-specific MLLMs is costly. This approach offers a reliable and scalable path to leverage foundation models in specialized domains, beyond medical data synthesis.

We are confident that these additions have made the paper stronger. We thank the reviewers and the Area Chair for their time and consideration.

---

### Decision · Program_Chairs · 2025-09-17

**Decision:**

Accept (poster)

**Comment:**

This work presents MAGIC, a novel framework that effectively generates clinically accurate synthetic dermatological images by fine-tuning diffusion models with AI-generated expert feedback from a multimodal LLM (GPT-4o). The paper's principal strength is its pragmatic and well-validated solution to a critical problem: reducing the immense burden of manual expert annotation in medical imaging. The proposed semi-automated workflow, which translates dermatologist checklists into machine-readable rewards for Direct Preference Optimization (DPO) or reward-based fine-tuning, is sound and demonstrates tangible, significant improvements in downstream diagnostic accuracy (e.g., +9.02% on a 20-class classification task). While reviewers noted weaknesses concerning the domain-transfer novelty, the inherent reliance on checklist quality, and the potential hallucinations of the MLLM judge, these are considerably outweighed by the method's practical utility, rigorous experimental design, and strong empirical results. The authors' rebuttal successfully addressed these concerns, as all reviewers upgraded their scores. The rebuttal provided sufficient justification and promised clarifications, thereby solidifying the paper's contribution to the field.